# Subfunctionalized expression drives evolutionary retention of ribosomal protein paralogs *Rps27* and *Rps27l* in vertebrates

**Adele Francis Xu[1,2], Rut Molinuevo[3], Elisa Fazzari[4,5,6], Harrison Tom[4,5,6], Zijian Zhang[7], Julien Menendez[3], Kerriann M Casey[8,9], Davide Ruggero[5,6], Lindsay Hinck[3], Jonathan K Pritchard[1], Maria Barna[1]***

[1]Department of Genetics, Stanford University, Stanford, United States; [2]Medical Scientist Training Program, Stanford School of Medicine, Stanford, United States; [3]Department of Molecular, Cell, and Developmental Biology, University of California, Santa Cruz, Santa Cruz, United States; [4]Helen Diller Family Comprehensive Cancer Center, University of California, Los Angeles, Los Angeles, United States; [5]Department of Cellular and Molecular Pharmacology, University of California, San Francisco, San Francisco, United States; [6]Department of Urology, University of California, San Francisco, San Francisco, United States; [7]Department of Chemical and Systems Biology, Stanford University, Stanford, United States; [8]Department of Biology, Stanford University, Stanford, United States; [9]Department of Comparative Medicine, Stanford School of Medicine, Stanford, United States

*For correspondence:
mbarna@stanford.edu

**Abstract** The formation of paralogs through gene duplication is a core evolutionary process. For paralogs that encode components of protein complexes such as the ribosome, a central question is whether they encode functionally distinct proteins or whether they exist to maintain appropriate total expression of equivalent proteins. Here, we systematically tested evolutionary models of paralog function using the ribosomal protein paralogs *Rps27* (*eS27*) and *Rps27l* (*eS27L*) as a case study. Evolutionary analysis suggests that *Rps27* and *Rps27l* likely arose during whole-genome duplication(s) in a common vertebrate ancestor. We show that *Rps27* and *Rps27l* have inversely correlated mRNA abundance across mouse cell types, with the highest *Rps27* in lymphocytes and the highest *Rps27l* in mammary alveolar cells and hepatocytes. By endogenously tagging the Rps27 and Rps27l proteins, we demonstrate that Rps27- and Rps27l-ribosomes associate preferentially with different transcripts. Furthermore, murine *Rps27* and *Rps27l* loss-of-function alleles are homozygous lethal at different developmental stages. However, strikingly, expressing Rps27 protein from the endogenous *Rps27l* locus or vice versa completely rescues loss-of-function lethality and yields mice with no detectable deficits. Together, these findings suggest that *Rps27* and *Rps27l* are evolutionarily retained because their subfunctionalized expression patterns render both genes necessary to achieve the requisite total expression of two equivalent proteins across cell types. Our work represents the most in-depth characterization of a mammalian ribosomal protein paralog to date and highlights the importance of considering both protein function and expression when investigating paralogs.

## Editor's evaluation

This article focuses on the fate of two ribosomal protein genes, *Rps27* and *Rps27L*, of vertebrates after they split in whole-genome duplication. The major strength is the support from solid laboratory experiments and an evolutionary perspective. It is a valuable case study revealing the differentiated roles in protein synthesis and expression patterns of duplicated genes.

## Introduction

Gene duplication is a fundamental evolutionary process that expands the complexity and functional repertoire of genomes (*Lynch and Conery, 2000*; *Nadeau and Sankoff, 1997*; *Ohno, 1970*). A variety of evolutionary models have been proposed to explain why some gene duplicates, or paralogs, become fixed in a population and conserved over long evolutionary time scales (*Conant and Wolfe, 2008*; *Innan and Kondrashov, 2010*; *Ohno, 1970*; *Prince and Pickett, 2002*). For some genes, an increase in copy number may be directly advantageous (beneficial dosage increase) (*Innan and Kondrashov, 2010*; *Kondrashov et al., 2002*; *Stark and Wahl, 1984*). In other cases, duplication creates redundancy that relaxes the purifying selection on both paralogs. Such conditions may allow one paralog to evolve regulatory features for expressing in new contexts (neofunctionalized expression) (*Force et al., 1999*; *Sidow, 1996*) or to encode a new gene product (i.e. a neofunctionalized protein) (*Ohno, 1970*). Rather than allow new functions, redundancy can alternatively allow paralogs to partially degenerate. If *cis*-regulatory features are included in a duplication, a newly formed paralog pair should initially exhibit symmetric expression. Under the relaxed selection, each may accumulate regulatory mutations of similar effect and continue expressing symmetrically at a lower total level, but often one paralog eventually degenerates more than the other (asymmetric expression) or undergoes nonfunctionalization altogether (*Lan and Pritchard, 2016*; *Lynch and Conery, 2000*). It is also possible for two paralogs' regulatory features to undergo complementary degeneration such that each becomes the major source of expression in a different subset of the ancestral contexts (subfunctionalized expression) (*Force et al., 1999*). Symmetric, moderately asymmetric, or subfunctionalized expression can promote retention of paralogs if they engage in dosage sharing, a situation in which both paralogs are needed to achieve the necessary total expression of the gene product (*Force et al., 1999*; *Lan and Pritchard, 2016*). Analogous to subfunctionalized expression, complementary degeneration of gene products (i.e. subfunctionalized proteins) can render both paralogs necessary to carry out the functions once performed by one ancestral gene product (*Hughes, 1994*). A final potential benefit of paralog retention is that one paralog may be able to compensate if the other paralog is disabled, thus conferring resilience against loss-of-function mutations (paralog buffering) (*De Kegel and Ryan, 2019*; *Gu et al., 2003*; *Thompson et al., 2021*).

Comparing theoretical models to experimental data on real-world paralogs is critical to understanding how genomes evolve and to predicting the effects of polymorphisms or therapeutic interventions involving paralogous genes. In terms of observable features in present-day paralogs, the above models fall into two categories: either the paralogs evolve to encode distinct gene products or the expression of both paralogs becomes advantageous or necessary even if they encode similar gene products. Importantly, classic knockout, knockdown, and overexpression experiments cannot conclusively distinguish whether observed effects are caused by perturbing a function specific to one paralog's gene product, or by altering total expression of two interchangeable gene products, or both. Instead, experimental designs that manipulate gene product characteristics without altering expression levels, or that isolate both gene products and their interactors from a source that expresses both paralogs, can help decouple the significance of expression level from that of divergent gene product characteristics. Editing an endogenous paralog gene to encode the gene product of the other paralog is an elegant approach of this nature that has occasionally been employed, as exemplified by previous work to substitute the exons of mouse *Hoxa3* for those of *Hoxd3* and vice versa (*Greer et al., 2000*), or to edit codons corresponding to differing residues between mouse actin paralogs (*Patrinostro et al., 2018*; *Vedula et al., 2017*). Rigorous experimental design was especially important in these cases because organisms are sensitive to the expression levels of Hox and actin genes (*Greer et al., 2000*; *Vedula et al., 2017*).

Sensitivity to expression level is also often encountered in genes encoding components of protein complexes, some of which require expression of all components at balanced dosages to achieve

effective assembly (*Papp et al., 2003*; *Taggart et al., 2020*). However, it is also noteworthy that the functions of protein complexes can be modulated by incorporating alternative isoforms of their components (*Antebi et al., 2017*; *Raices and D'Angelo, 2012*). It is thus especially important to determine whether paralogs of protein complex components serve as necessary sources of expression, as functionally distinct proteins, or both. In particular, growing interest surrounds the paralogs of genes that encode the protein components of the ribosome, the ubiquitous macromolecular complex that catalyzes translation of all protein-coding transcripts (*Genuth and Barna, 2018*; *Gerst, 2018*; *Komili et al., 2007*; *Topisirovic and Sonenberg, 2011*). In mammals, each ribosome consists of 80 ribosomal proteins (RPs) and 4 ribosomal RNAs (rRNAs), which together form a 40S small subunit and a 60S large subunit that join during translation (*Anger et al., 2013*; *Uechi et al., 2001*). Each RP is encoded by a different genomic locus. Ten of these RP genes are known to exist as paralog pairs in the human genome, several of which are conserved among other mammals (*Balasubramanian et al., 2009*; *Gupta and Warner, 2014*; *Nakao et al., 2004*; *Sugihara et al., 2010*).

Two main categories of hypotheses could explain why some duplicated RP genes have been evolutionarily conserved. The first category posits that RP paralogs have diverged in their protein functions: they may have acquired distinct extraribosomal functions, which have been observed previously for certain RPs (*Warner and McIntosh, 2009*; *Zhang et al., 2017b*), or they may encode alternative RP isoforms that assemble to form ribosomes with different translational characteristics (*Gerst, 2018*). The latter possibility is especially intriguing, given that other variations in ribosome composition have been observed to affect protein synthesis or modulate the translation of specific genes (*Genuth and Barna, 2018*; *Xue and Barna, 2012*). Studies in yeast, in which 59 of the 79 RP genes exist as paralog pairs, have motivated speculation that functionally distinct RP paralogs are a major component of translational gene regulation through the formation of ribosomes that are heterogeneous in composition, a model known as the 'ribosome code' (*Ghulam et al., 2020*; *Komili et al., 2007*; *Parenteau et al., 2015*). Conversely, one could propose a second category of hypotheses that RP paralogs do not encode functionally distinct proteins, but are rather retained because their expression has become necessary. This possibility would be consistent with observations suggesting that, for paralogs in general, retention due to expression may be more common than divergence of protein function (*Prince and Pickett, 2002*). Such observations include systems-level analysis showing that paralogs found throughout fungal genomes rarely change gene ontologies or protein interaction networks (*Wapinski et al., 2007*), and experimental demonstrations for individual genes such as Hox paralogs that suggest equivalent protein function despite distinct expression patterns (*Bruce et al., 2001*; *Greer et al., 2000*).

In this work, we systematically examine these two categories of hypotheses in the case of the RP paralogs *Rps27* and *Rps27l* (known in standardized RP nomenclature as *eS27* and *eS27L*, respectively *Ban et al., 2014*). Previous work has shown that both *Rps27* and *Rps27l* are expressed in most tissues and are incorporated into actively translating ribosomes (*O'Donohue et al., 2010*; *Xiong et al., 2014*). While no *Rps27* knockout mouse has previously been described, homozygous *Rps27l* knockout is lethal at early postnatal stages with increased apoptosis in hematopoietic organs (*Xiong et al., 2014*). *Rps27* and *Rps27l* are differentially regulated upon activation of the tumor suppressor *Trp53* and have opposite feedback effects on the *Trp53-Mdm2* axis (*He and Sun, 2007*; *Xiong et al., 2014*; *Xiong et al., 2011*). However, the study of *Rps27* and *Rps27l* is complicated by the consideration that impaired ribosome biogenesis due to perturbed dosage balance of other RPs has also been shown to activate *Trp53* via nucleolar stress or translation inhibition (*Nicolas et al., 2016*; *Russo and Russo, 2017*; *Zhang and Lu, 2009*). Thus, the knockout, knockdown, and overexpression experiments that have primarily been used to date could either reflect *Rps27* and *Rps27l*'s direct role in *Trp53* signaling or the nonspecific effects of perturbing ribosome biogenesis.

The challenges of interpreting the existing literature on *Rps27* and *Rps27l* informed our approach to investigating these paralogs without perturbing their expression. To provide the context of evolutionary history, we first examined the copy number and molecular phylogeny of *Rps27* and *Rps27l* across representative animal species and the intraspecies synteny between the *Rps27* and *Rps27l* loci, thereby corroborating the origin of this paralog pair via an ancient whole-genome duplication. Next, we examined single-cell transcriptomics of normal mouse tissues and found previously unreported cell type-specific patterns of *Rps27* and *Rps27l* expression. We confirmed that the Rps27 and Rps27l proteins are incorporated into ribosomes and used endogenously epitope-tagged Rps27 and Rps27l

combined with immunoprecipitation and ribosome profiling to examine whether Rps27l- and Rps27l-ribosomes associate with different transcripts. We then examined the functions of *Rps27* and *Rps27l* at an organismal level by first generating loss-of-function alleles for each gene and, most importantly, generating 'homogenized' mouse lines in which the *Rps27* locus has been edited to encode the Rps27l protein, or vice versa. Finally, we performed a detailed characterization of the homogenized mouse lines with particular attention to the cell types that preferentially express one paralog and also examined the effects of paralog homogenization on aging and response to genotoxic stress.

In sum, investigating the functions of RP paralogs is valuable for elucidating the evolutionary fates of paralogs that encode components of protein complexes and may shed light on the potential role of paralogs in the ribosome code. Such work demands careful distinctions between the possibility that paralogs encode functionally distinct proteins, as opposed to the possibility that they provide expression of similar proteins. With close attention to these challenges, we present here the most in-depth profiling of a mammalian RP paralog to date, in which we have leveraged single-cell transcriptomics, molecular assays, and mouse genetics to comprehensively examine the above hypotheses regarding paralog function in the case of *Rps27* and *Rps27l*.

## Results

### *Rps27* and *Rps27l* are vertebrate ohnologs encoding highly conserved proteins

To guide our functional analysis of *Rps27* and *Rps27l*, we first considered their probable evolutionary trajectory by examining their gene structure, intraspecies synteny, and interspecies molecular phylogeny. In general, several lines of evidence can support whether a duplicated gene arose through DNA-based events such as DNA transposition, tandem duplication, segmental duplication, or whole-genome duplication (WGD); or through retrotransposition of RNA (reviewed in *Graur and Li, 1997*). In human and mouse, *Rps27* and *Rps27l* are located on separate chromosomes. Unlike the approximately 2000 mostly intronless processed RP pseudogenes resulting from retrotransposition in these genomes (*Balasubramanian et al., 2009*), the canonical transcripts for *Rps27* and *Rps27l* each contain three introns and their exon junctions correspond to similar positions within the encoded proteins (*Figure 1—figure supplement 1A*). Furthermore, the *Rps27l* locus is flanked by several other genes that also have paralogs near the *Rps27* locus and are arranged in similar order (*Figure 1A*). Thus, these syntenic genomic regions likely originated via DNA-based duplication.

To infer the evolutionary timing of this duplication, we examined the copy number and molecular phylogeny of *Rps27* and *Rps27* across genomes from invertebrates, jawless vertebrates, cartilaginous and bony fish, amphibians, sauropsids, and mammals (*Supplementary file 1*). While none of the nine included invertebrates had more than one *Rps27* ortholog, nearly all vertebrates had two *Rps27* paralogs, and most teleost fish species had three to six (*Figure 1B*). Additionally, in a phylogenetic tree constructed from the coding sequences of *Rps27* and *Rps27l* across species, the *Rps27* sequences from mammals and coelacanth formed a distinct clade from the corresponding *Rps27l* sequences (*Figure 1—figure supplement 1B*), suggesting that duplication and subsequent divergence of these loci began in a common ancestor of these species. Interestingly, it has long been hypothesized that at least one WGD occurred in a common ancestor of all vertebrates ('1R'), followed closely by either a second WGD ('2R') with subsequent loss of many duplicates or by several large segmental duplications (*Dehal and Boore, 2005*; *Nakatani et al., 2021*; *Ohno, 1970*; *Sacerdot et al., 2018*; *Simakov et al., 2020*; *Smith et al., 2018*). Indeed, in systematic efforts to map 1R/2R remnants throughout vertebrate genomes based on phylogeny and large-scale synteny (*Makino and McLysaght, 2010*; *Sacerdot et al., 2018*; *Singh and Isambert, 2020*), *Rps27* and *Rps27l* are consistently identified among the 'ohnolog' gene duplicates that still comprise 25–35% of present-day vertebrate genes. Furthermore, a third WGD occurred in the common ancestor to teleost fish. Carp and salmon, which each have six *Rps27* paralogs, have each experienced an independent fourth WGD (*Macqueen and Johnston, 2014*; *Xu et al., 2019*). Thus, increases in the number of *Rps27* paralogs coincide in evolutionary timing with WGDs.

The probable evolutionary trajectory of *Rps27* and *Rps27l* should also be considered in relation to other RPs: in fungi, genes encoding protein complexes such as the ribosome often remain duplicated after WGD but rarely duplicate on an individual basis. This observation supports a model that assumes

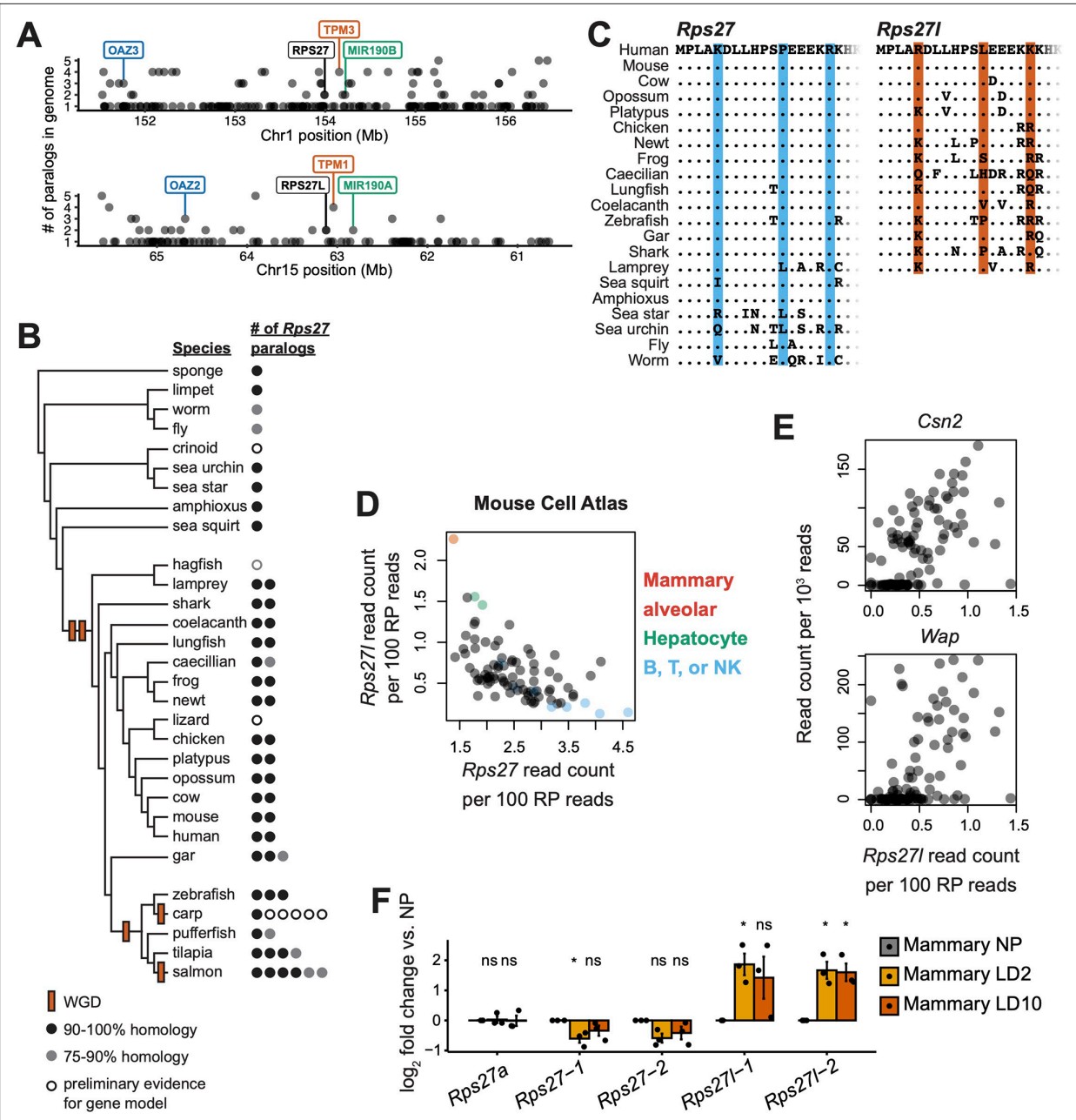

**Figure 1.** *Rps27* and *Rps27l* evolutionary origin and cell type-specific mRNA abundance. (**A**) 5 Mb windows centered on the *Rps27* and *Rps27l* loci in human genome (GRCh38.p13). Points indicate individual Ensembl-annotated (version 109) genes. Y-axis indicates total number of Ensembl-annotated paralogs in the genome for each gene. Label colors indicate paralogous genes. (**B**) Phylogenetic tree of representative animal species showing putative timing of whole-genome duplications (WGD) and number of *Rps27* paralogs per species. See details for included genes in ***Supplementary file 1***. (**C**) Multi-species alignment of Rps27 and Rps27l N-terminal protein sequences. For species with >2 *Rps27* paralogs, the protein sequences with the highest homology to human Rps27 and Rps27l are shown. The three residues that differ between Rps27 and Rps27l in human and mouse are shaded. (**D**) *Rps27* and *Rps27l* scRNA-seq values across cell types from the Mouse Cell Atlas (***Han et al., 2018***; Pearson's r = –0.58, p=4.4e-09). See also ***Figure 1—figure supplement 2***. (**E**) Correlation between *Rps27l* and milk protein transcripts in scRNA-seq (***Bach et al., 2017***) from alveolar cells in lactating mammary glands ('Avd-L' cells as termed by Bach et al.). *Csn2*: Spearman's $\rho$ = 0.63, p=5.7e-13. *Wap*: Spearman's $\rho$ = 0.56, p=6.7e-10. (**F**) RT-qPCR of *Rps27*, *Rps27l*, and a control RP gene, *Rps27a*, in mammary glands of nulliparous and lactating female mice. Values are normalized to *Rps6* and are shown as log fold differences over NP. For *Rps27* and *Rps27l*, two independent primer sets ('–1' and '–2') were used. n = 3 biological replicates. Significance compared to NP was assessed by t-test. *Rps27*-1 LD2: p=0.050; *Rps27l*-1 LD2: p=0.035; *Rps27l*-2 LD2: p=0.028; *Rps27l*-2 LD10: p=0.032. 'ns' indicates p>0.05. Error bars show standard error.

The online version of this article includes the following figure supplement(s) for figure 1:

*Figure 1 continued on next page*

*Figure 1 continued*

**Figure supplement 1.** Supplementary cross-species comparison of *Rps27* and *Rps27l*.

**Figure supplement 2.** Supplementary cell type-dependent expression data for *Rps27* and *Rps27l*.

dosage balance is required between components of a protein complex, such that duplicating one gene would cause a deleterious imbalance but simultaneously duplicating all the components would maintain balance and favor initial retention of the resulting paralogs (*Papp et al., 2003*; *Wapinski et al., 2007*). Notably, most RPs in present-day mammalian genomes are encoded by a single gene; there are only 10 known human RP paralog pairs, several of which have hallmarks of tandem duplication or retrotransposition rather than WGD (*Gupta and Warner, 2014*). Two evolutionary trajectories are thus possible: either all RP genes were duplicated via WGD and *Rps27* is one of the few that did not eventually revert to a single gene through nonfunctionalization of one duplicate, or a smaller-scale duplication occurred for *Rps27* alone and survived despite initial excess dosage. Based on the above evidence that *Rps27* and *Rps27l* are vertebrate-specific, we consider 1R/2R WGD to be the more parsimonious explanation for their origin.

Based on these evolutionary features of *Rps27* and *Rps27l*, we can preliminarily assess the relevance of the proposed paralog retention models. Given the evidence from fungi that RP genes are subject to dosage balance constraints, beneficial dosage increase and neofunctionalized expression are less likely models of paralog retention in this case. However, the other proposed models all remain plausible. In contrast to tandem duplications in which both paralogs reside near the same regulatory features (*Lan and Pritchard, 2016*) or transpositions that introduce a duplicate into a completely different regulatory context, WGD produces duplicates on separate chromosomes with initially identical regulatory features that may then diverge. Notably, while the coding sequences of *Rps27* and *Rps27l* are highly divergent, almost all substitutions are synonymous and their protein sequences only differ by three N-terminal residues out of a total of 84: K5R, P12L, and R17K. These differences between the two paralogous proteins are well-conserved among mammals (*Figure 1C*). This high degree of protein sequence conservation over a long evolutionary history suggests that the proteins still perform related molecular roles in the cell and could thus participate in dosage sharing or paralog buffering. On the other hand, the three differing residues could confer partially distinct protein functions by affecting protein structure or post-translational modifications. Thus, we proceeded to compare both the expression patterns and the protein characteristics of *Rps27* and *Rps27l*.

### *Rps27* and *Rps27l* mRNA expression are cell type-dependent

We next leveraged publicly available single-cell RNA-seq datasets to compare expression levels of *Rps27*, *Rps27l*, and other RP genes in previously unexamined primary cell types. Different cell types express RP genes at different levels, likely reflecting the ribosome production rate needed to accommodate each cell type's translational or proliferative demands. However, transcript abundance among the core RP genes is highly correlated across cell types (*Figure 1—figure supplement 2A*), which is consistent with the theory that components of a protein complex must have balanced expression for effective assembly (*Guimaraes and Zavolan, 2016*; *Papp et al., 2003*). Thus, for each cell, we used the summed transcript abundances of all RP genes to normalize each RP gene's expression level by the cell type's ribosome production rate, allowing us to identify RP genes with disproportionately high or low expression relative to other RP genes in a cell type.

By analyzing single-cell data from annotated cell types across dozens of mouse tissues in the Mouse Cell Atlas (*Han et al., 2018*), we first were able to corroborate the previously reported tissue-specific expression patterns of several other RP paralogs (*Figure 1—figure supplement 2B*; *Chaillou et al., 2016*; *Guimaraes and Zavolan, 2016*; *Jiang et al., 2017*; *Sugihara et al., 2010*; *Wong et al., 2014*). Interestingly, we also found that *Rps27* and *Rps27l* mRNA levels are inversely correlated across cell types (*Figure 1D*). The highest *Rps27l:Rps27* ratios were found in mammary alveolar cells and hepatocytes; the lowest were found in a subset of B, T, and NK lymphocytes. A similar expression pattern was detected among FACS-isolated hepatocytes and lymphocytes collected in Tabula Muris (*Tabula Muris Consortium, 2018*), a separately constructed large dataset of mouse single-cell RNA-seq samples (*Figure 1—figure supplement 2C*).

Mammary alveolar cells are a transient cell type that arise rapidly from luminal progenitor cells in female mammary glands during pregnancy and lactation to secrete milk components (*Macias and*

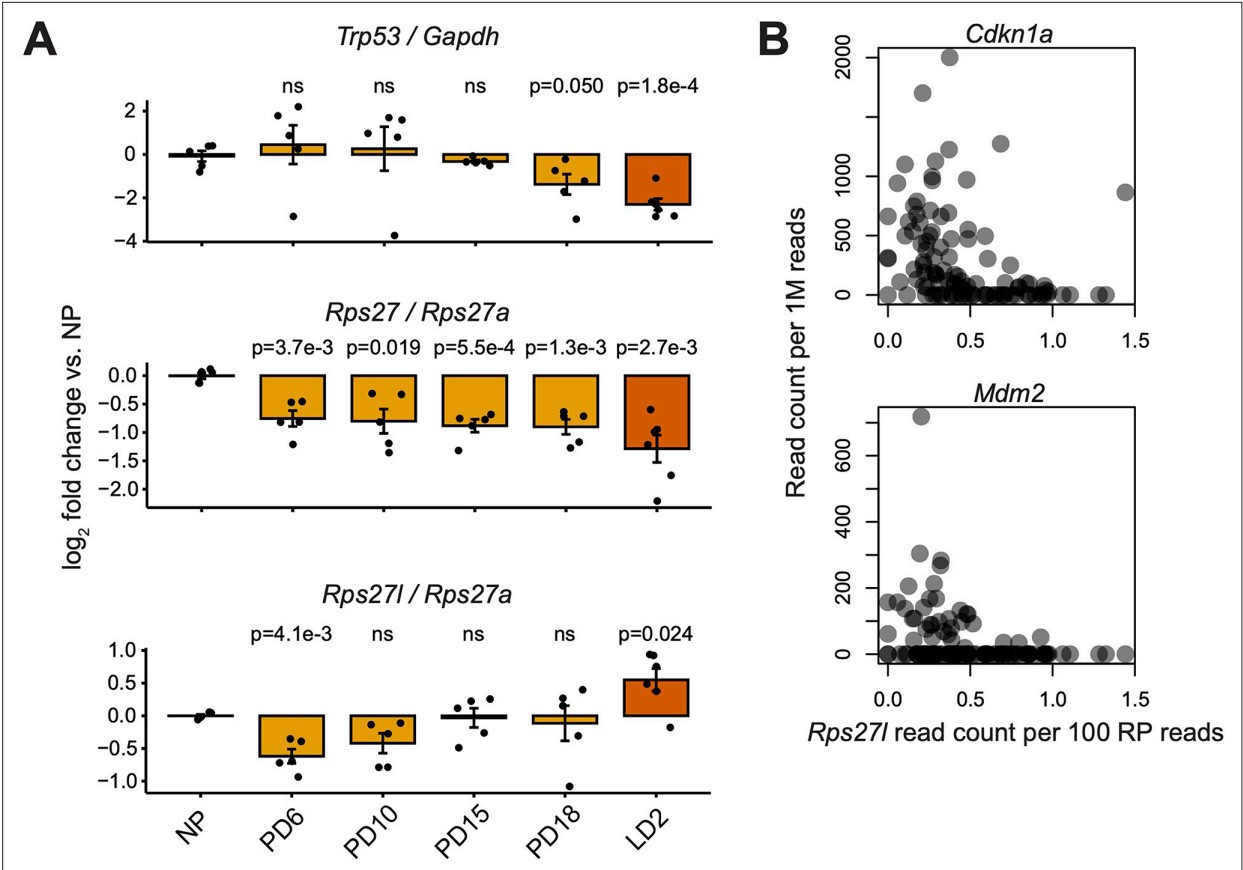

**Figure 2.** Comparison of *Rps27* and *Rps27l* mRNA abundance versus *Trp53* expression and activity. (**A**) RT-qPCR of *Rps27*, *Rps27l*, and *Trp53* in mammary glands at nulliparous (NP), pregnancy days 6–18 (PD6–18), and lactation day 2 (LD2) timepoints. *Rps27* and *Rps27l* are normalized by ribosomal protein (RP) *Rps27a*. *Trp53* is normalized by *Gapdh*. n = 5–6 biological replicates (individual animals) per timepoint. Significance versus NP was assessed by *t*-test. 'ns' indicates p>0.05. Error bars show standard error. (**B**) Correlation between *Rps27l* and other targets of *Trp53* transcriptional activation in scRNA-seq of alveolar cells from lactating mammary glands (***Bach et al., 2017***). *Cdkn1a*: Spearman's $\rho$ = –0.51, p=3.7e-8. *Mdm2*: Spearman's $\rho$ = –0.37, p=1.2e-4.

*Hinck, 2012*). Because lactating mammary glands were not collected in Tabula Muris, we corroborated the Mouse Cell Atlas findings by analyzing a single-cell RNA-seq dataset of mammary development timepoints (***Bach et al., 2017***), and a bulk RNA-seq dataset of sorted mouse mammary cell types (***Fu et al., 2015***). We found that *Rps27l* mRNA levels increase in luminal cells between pregnancy and lactation, while *Rps27* mRNA levels diminish (***Figure 1—figure supplement 2D***). Among mature alveolar cells, single cells expressing the highest levels of milk protein transcripts such as *Csn2* and *Wap* also had the highest *Rps27l* expression (***Figure 1E***). Indeed, these transcripts correlated more strongly with *Rps27l* expression than almost all others in the transcriptome (***Supplementary file 2***). Notably, these datasets were collected from mice of different strains, suggesting a generalizable trend.

To confirm these in silico findings, we performed RT-qPCR for *Rps27* and *Rps27l* on bulk mouse mammary tissue from nulliparous and lactating females (***Figure 1F***). Liver, heart, and brain samples from the same animals were also analyzed (***Figure 1—figure supplement 2E***). Liver and lactating mammary gland had higher levels of *Rps27l* and lower levels of *Rps27* than nulliparous mammary gland, heart, or brain. Ribosomal protein *Rps6* (eS6) was used to normalize for the ribosome production rate of each tissue. Ribosomal protein *Rps27a* (eS31) was included as an example of an RP whose transcript abundance does not change during lactation when normalized by *Rps6*.

It has previously been reported that *Rps27l* is transcriptionally upregulated by *Trp53* via two Trp53-binding elements within the *Rps27l* genomic locus, and that *Trp53* also downregulates *Rps27* expression (***He and Sun, 2007***; ***Li et al., 2007***; ***Xiong et al., 2014***; ***Xiong et al., 2011***). To determine whether *Trp53* activity and *Rps27* or *Rps27l* expression may be correlated in the mammary gland,

we performed RT-qPCR for *Trp53*, *Rps27*, *Rps27l*, and other RP transcripts on bulk mammary tissue at nulliparous, pregnant, and lactating timepoints (*Figure 2A*). *Trp53* transcript abundance is stable during early-to-mid pregnancy, whereas *Rps27* and *Rps27l* both decrease within the first 6 days of pregnancy. *Trp53* transcript abundance then decreases in the latter half of pregnancy, whereas *Rps27l* increases and *Rps27* remains decreased around lactation day 2 (LD2). Recognizing that *Trp53* activity can be modulated through post-transcriptional mechanisms not reflected by *Trp53* mRNA abundance (*Ashcroft and Vousden, 1999*; *Meek and Anderson, 2009*), we again used a single-cell RNA-seq dataset of mammary gland timepoints (*Bach et al., 2017*) to compare *Rps27l* expression against the expression of other genes that are transcriptionally activated by *Trp53* (*Figure 2B*). Among differentiated mammary alveolar cells, we found that *Rps27l* expression is inversely correlated with mRNA abundance of other *Trp53*-upregulated genes, suggesting that single cells with high *Rps27l* expression do not have increased *Trp53* activity. These findings, therefore, suggest that differences in *Rps27* and *Rps27l* expression across cell types are not primarily driven by *Trp53* activity.

Our findings that *Rps27* and *Rps27l* have inversely correlated mRNA abundance across cell types suggest that *Rps27* and *Rps27l* have complementary expression patterns, a hallmark of subfunctionalized expression. While *Rps27* and *Rps27l* have previously been included in bulk transcriptomic analysis of RPs (*Guimaraes and Zavolan, 2016*; *Gupta and Warner, 2014*; *He and Sun, 2007*), our observations were facilitated by including lactating mammary gland samples and achieving single-cell resolution of cell types. Importantly, dosage sharing via subfunctionalized expression could alone explain the retention of these paralogs through a duplication–degeneration–complementation (DDC) process (*Force et al., 1999*): whereas the ancestral *Rps27* gene may have been expressed widely, the two paralogs may have lost complementary sets of regulatory elements until both were needed to maintain the appropriate expression level across all cell types. If *Rps27* and *Rps27l* are indeed remnants of WGD as discussed in the previous section, DDC of *Rps27* and *Rps27l* may have happened in concert with the other RP paralog pairs' reversion to single genes.

## Rps27 and Rps27l-ribosomes differentially associate with cell cycle-related mRNAs

Having established their distinct expression patterns, it remained to be explored whether the two paralogs could not only have different expression but also encode functionally different proteins, which has been observed with other paralogs (*Conant and Wolfe, 2006*). Using mouse embryonic stem cells (mESCs) as a primary cell line amenable to genetic editing, we first confirmed whether Rps27 and Rps27l are incorporated into actively translating ribosomes. mESC lysate was fractionated over a sucrose density gradient to separate mRNAs based on the number of ribosomes bound to them. Using antibodies specific to each paralog with only trace cross-reactivity to the other paralog (*Figure 3—figure supplement 1A*), we found that Rps27 and Rps27l are both detectable among fractions corresponding to actively translating ribosomes (polysomes), fractions corresponding to single ribosomes (80S), and fractions representing the 40S small subunit. Rps27 and Rps27l are not detectable in early fractions where most extra-ribosomal proteins are found (*Figure 3A*). In the structure of the human ribosome (*Natchiar et al., 2017*), Rps27 is incorporated into the 40S subunit such that the three paralog-specific amino acid positions are solvent-accessible (*Figure 3B*) and could modulate interactions with other molecules.

To compare the molecular interactions of Rps27- and Rps27l-ribosomes, we devised a strategy to isolate the two ribosome populations from in vitro cultured mouse cells and focused on comparing the mRNAs with which they associate. To enable isolation of Rps27- and Rps27l-ribosomes without overexpressing an exogenous epitope-tagged construct, we used CRISPR to homozygously insert 3xFLAG epitope tags immediately preceding the stop codon at the *Rps27* and *Rps27l* loci (*Figure 3C*). This approach yielded three independently selected clones (biological replicates) of *Rps27-FLAG* mESCs and three of *Rps27l-FLAG* mESCs. Rps27- and Rps27l-FLAG proteins are expressed at comparable levels to the untagged proteins and are incorporated into actively translating ribosomes (*Figure 3—figure supplement 1B and C*). We confirmed by western blot that anti-FLAG immunoprecipitation (FLAG-IP) efficiently enriches for Rps27- and Rps27l-ribosomes from the *Rps27-FLAG* and *Rps27l-FLAG* mESCs, respectively (*Figure 3—figure supplement 1D*). Minimal Rps27 is detected when targeting Rps27l for pulldown and vice versa, confirming that a ribosome does not simultaneously contain Rps27 and Rps27l and that we did not isolate undigested polysomes containing multiple

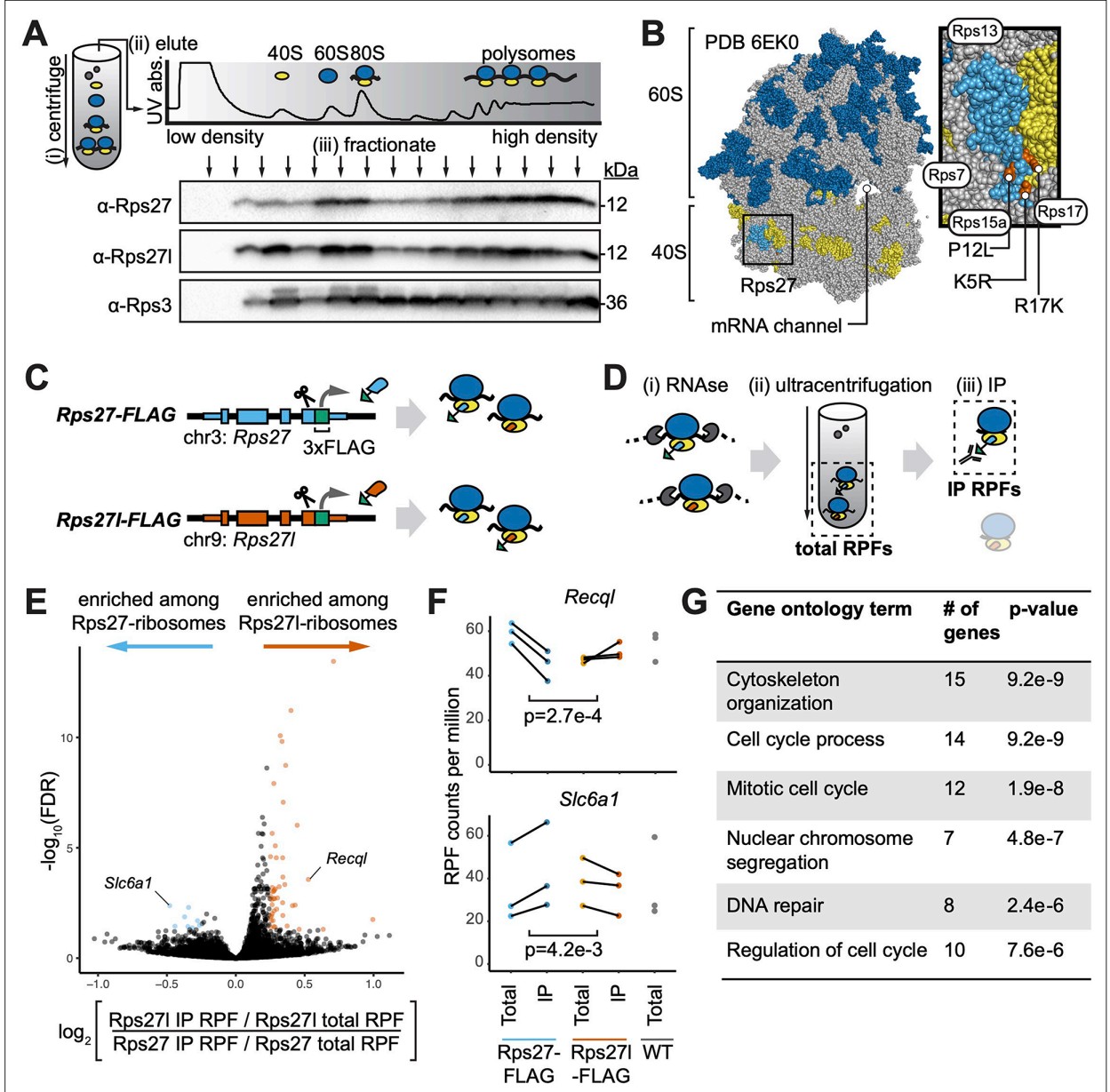

**Figure 3.** Paralog-specific ribosome profiling reveals preferential association of Rps27l-ribosomes with cell cycle-related mRNAs. (**A**) Fractionation of mouse embryonic stem cell (mESC) lysate by density to separate non-ribosomal proteins, ribosome 40S and 60S subunits, mRNAs bound by one ribosome (80S), and mRNAs bound by multiple ribosomes (polysomes). See *Figure 3—source data 1–3*. (**B**) Ribosome structure showing the 60S subunit (with 28S rRNA in dark blue), 40S subunit (with 18S rRNA in yellow), and Rps27 protein (light blue) within the 40S. Orange denotes residues that differ between Rps27 and Rps27l. Gray denotes other RPs. Neighboring RPs are labeled. (**C**) CRISPR-mediated insertion of 3xFLAG epitope tag C-terminally at the endogenous *Rps27* and *Rps27l* genomic loci in mESCs. (**D**) FLAG-IP ribosome profiling of paralog-containing ribosomes. (i) Cell lysate is treated with RNAse to digest mRNA sequences that are not protected by bound ribosomes. (ii) Ribosomes and associated ribosome-protected fragments (RPFs) are collected by ultracentrifugation. These are termed total RPFs. (iii) anti-FLAG immunoprecipitation (IP) is performed on total ribosomes. IP RPFs are eluted. (**E**) Comparison of RPFs enriched after IP in *Rps27l-FLAG* versus *Rps27-FLAG* mESCs. (**F**) Examples of genes differentially enriched upon Rps27-FLAG or Rps27l-FLAG pulldown. Significance was assessed using an empirical Bayes method for detecting differential expression, applied to a multilevel linear model to find genes whose RPF abundance is differentially affected by IP from *Rps27-* and *Rps27l-FLAG* mESCs. n = 3 biological replicates (independent mESC clones). Multiple-hypothesis-corrected false discovery rates (FDRs) are shown. (**G**) Top enriched gene ontology terms for RPFs that preferentially associate with Rps27l-ribosomes. See also *Figure 3—figure supplement 1* and *Appendix 1—figure 1*.

The online version of this article includes the following source data and figure supplement(s) for figure 3:

**Source data 1.** Rps27 density fractionation western blot.

**Source data 2.** Rps27l density fractionation western blot.

*Figure 3 continued on next page*

*Figure 3 continued*

**Source data 3.** Rps3 density fractionation western blot.

**Source data 4.** Editable data from *Figure 3G*.

**Figure supplement 1.** Supplementary *Rps27* and *Rps27l* epitope tagging, immunoprecipitation, and ribosome profiling data.

**Figure supplement 1—source data 1.** Rps27-GFP-FLAG and Rps27l-GFP-FLAG, FLAG western blot.

**Figure supplement 1—source data 2.** Rps27-GFP-FLAG and Rps27l-GFP-FLAG, Rps27 western blot.

**Figure supplement 1—source data 3.** Rps27-GFP-FLAG and Rps27l-GFP-FLAG, Rps27l western blot.

**Figure supplement 1—source data 4.** *Rps27-FLAG* and *Rps27l-FLAG* mouse embryonic stem cells (mESCs), Rps27 western blot.

**Figure supplement 1—source data 5.** *Rps27-FLAG* and *Rps27l-FLAG* mouse embryonic stem cells (mESCs), Rps27l western blot.

**Figure supplement 1—source data 6.** *Rps27-FLAG* and *Rps27l-FLAG* mouse embryonic stem cells (mESCs), FLAG western blot.

**Figure supplement 1—source data 7.** *Rps27-FLAG* and *Rps27l-FLAG* mouse embryonic stem cells (mESCs), beta-actin western blot.

**Figure supplement 1—source data 8.** WT mouse embryonic stem cell (mESC) density fractionation, FLAG western blot.

**Figure supplement 1—source data 9.** *Rps27-FLAG* mouse embryonic stem cell (mESC) density fractionation, FLAG western blot.

**Figure supplement 1—source data 10.** *Rps27l-FLAG* mouse embryonic stem cell (mESC) density fractionation, FLAG western blot.

**Figure supplement 1—source data 11.** *Rps27-FLAG* and *Rps27l-FLAG* mouse embryonic stem cell (mESC) FLAG-IP, Rps27 western blot.

**Figure supplement 1—source data 12.** *Rps27-FLAG* and *Rps27l-FLAG* mouse embryonic stem cell (mESC) FLAG-IP, Rps27l western blot.

**Figure supplement 1—source data 13.** *Rps27-FLAG* and *Rps27l-FLAG* mouse embryonic stem cell (mESC) FLAG-IP, Rps3 western blot.

**Figure supplement 1—source data 14.** *Rps27-FLAG* and *Rps27l-FLAG* mouse embryonic stem cell (mESC) FLAG-IP, Gapdh western blot.

ribosomes (*Figure 3—figure supplement 1D*). We then performed ribosome profiling on the total ribosome population in each line and on the paralog-containing ribosomes isolated via FLAG-IP (*Figure 3D*). Ribosome profiling identifies mRNA regions occupied by ribosomes, using RNAse digestion to enable sequencing of ribosome-protected fragments of mRNA (RPFs) (*Ingolia et al., 2009*). The total RPFs from the two FLAG-tagged cell lines did not contain mRNAs that differed significantly in abundance either in comparison between the *Rps27-FLAG* line and *Rps27l-FLAG* line or in comparison to a passage-matched wild-type (WT) control line (*Figure 3—figure supplement 1E*). This confirms that FLAG-tagging did not alter the landscape of normally translated mRNAs. We then compared the IP RPFs from both cell lines to each other using the total RPFs from each clone to normalize for its overall translational landscape (*Figure 3E and F*). After excluding a *Tmod3* transcript as a likely artifact of FLAG-IP (Appendix 1, *Appendix 1—figure 1*), we identified 8 transcripts enriched among Rps27-ribosomes and 46 transcripts enriched among Rps27l-ribosomes (absolute value of $\log_2$ fold change > 0.25, false discovery rate < 0.05; *Supplementary file 3*). Among the latter set of transcripts, Gene Ontology terms associated with cell cycle processes were enriched (*Figure 3G*). These findings intriguingly demonstrated that Rps27- and Rps27l-ribosomes associate differently with specific mRNAs.

Importantly, our experimental design minimized the possibility that these results reflect translational changes due to genetically editing the cell lines: no knockdown, knockouts, or overexpression were used, and the IP RPF transcript abundances were normalized by the total RPF transcript abundances for each clone. A critical consideration, however, is that this approach demonstrates a correlation between Rps27l incorporation and ribosome association with cell cycle-related transcripts, but it does not directly demonstrate that Rps27l causes ribosomes to preferentially bind these transcripts. To definitively compare the functions of the two proteins, we next turned to in vivo approaches.

## *Rps27* and *Rps27l* homozygous knockouts are lethal at different developmental stages

To compare *Rps27* and *Rps27l* in vivo, we first examined the organism-level effects of knocking out each paralog. We generated mice harboring two truncation alleles at the endogenous *Rps27* and *Rps27l* loci: *Rps27^exon2del^*, in which the splicing junctions flanking exon 2 of *Rps27* are deleted; and *Rps27l^exon2del^*, which harbors a 320 bp deletion in *Rps27l* that spans exon 2 and part of the subsequent intron (*Figure 4A*). Similar to a previously described *Rps27l* gene-trapped loss-of-function mouse model (here termed *Rps27l^GT^*) (*Xiong et al., 2014*), *Rps27l^exon2del / +^* males and females are viable and fertile, but *Rps27l^exon2del / exon2del^* mice are observed at lower-than-expected frequencies in crosses of

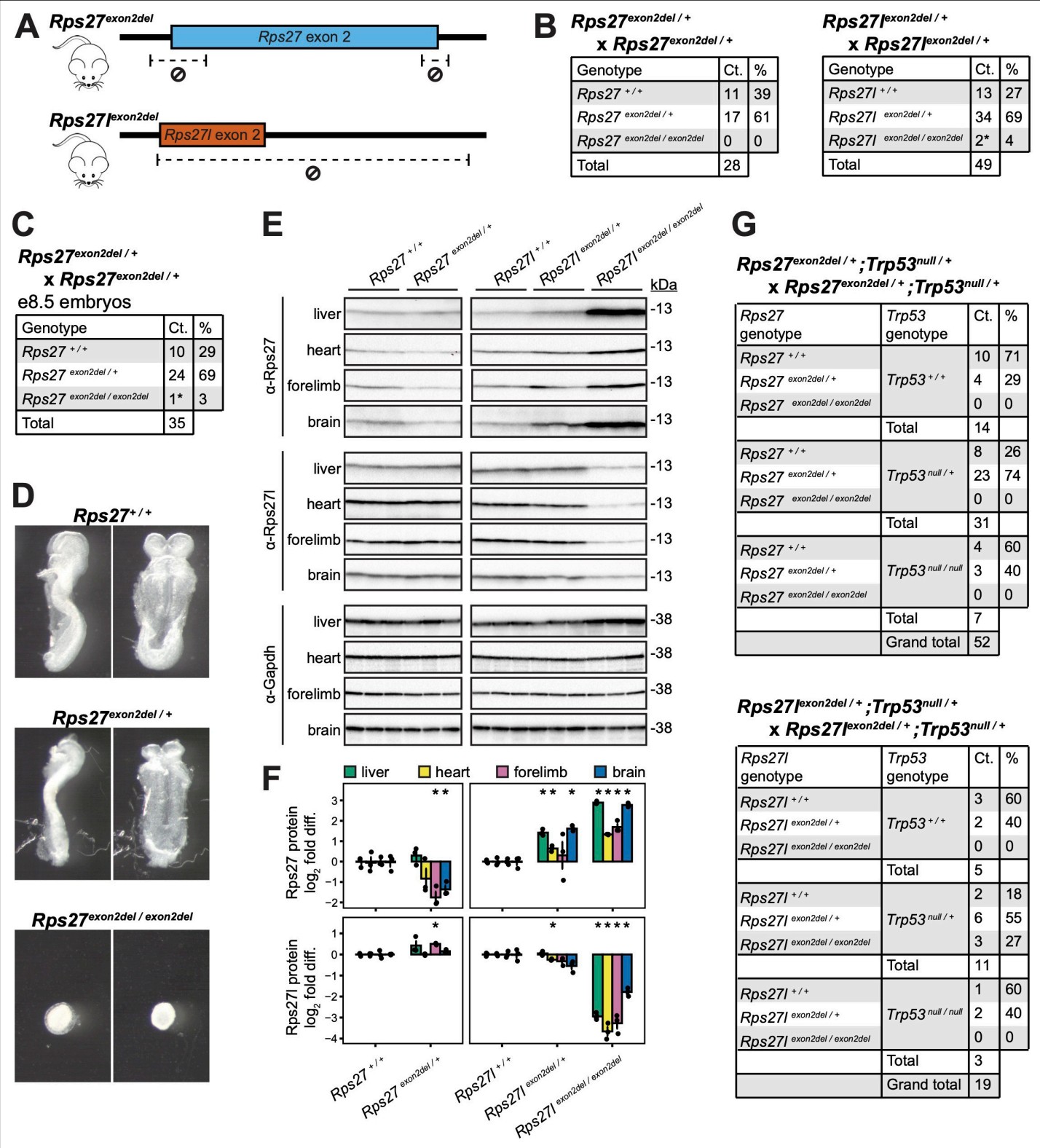

**Figure 4.** *Rps27* and *Rps27l* truncation alleles are homozygous lethal at embryonic and early postnatal stages, respectively. (**A**) CRISPR-mediated deletions in *Rps27^exon2del^* and *Rps27l^exon2del^* alleles. (**B**) Genotype ratios among live offspring of *Rps27^exon2del / +^* and *Rps27l^exon2del / +^* heterozygous crosses at postnatal day 13 (P13). * indicates animals that died or required euthanasia by postnatal days 13–17 (P13–17). (**C**) Genotype ratios and (**D**) photographs of embryonic day 8.5 (e8.5) embryos from *Rps27^exon2del / +^* heterozygous crosses. * indicates severely delayed development. (**E**) Western blots of postnatal day 0 (P0) tissues with Gapdh as loading control and n = 3 biological replicates (animals) per genotype. The early embryonic lethal *Rps27^exon2del / exon2del^*

*Figure 4 continued on next page*

*Figure 4 continued*

genotype was excluded due to lack of sufficient obtainable tissue. (**F**) Rps27 and Rps27l protein abundance in (**E**) quantified as Gapdh-normalized log$_2$ fold difference relative to averaged +/+ samples for each tissue. Error bars show standard error. *p<0.05 by *t*-test; unmarked = not significant. (**G**) Genotype ratios among live offspring of *Rps27*$^{exon2del / +}$;*Trp53*$^{null / +}$ double heterozygous crosses and *Rps27l*$^{exon2del / +}$;*Trp53*$^{null / +}$ double heterozygous crosses at P13. See ***Figure 4—source data 1–15***.

The online version of this article includes the following source data for figure 4:

**Source data 1.** *Rps27*$^{exon2del}$ P0 liver Rps27 western blot.

**Source data 2.** *Rps27l*$^{exon2del}$ P0 liver Rps27 western blot.

**Source data 3.** *Rps27*$^{exon2del}$ P0 heart Rps27 western blot.

**Source data 4.** *Rps27l*$^{exon2del}$ P0 heart Rps27 western blot.

**Source data 5.** *Rps27*$^{exon2del}$, *Rps27l*$^{exon2del}$ P0 forelimb Rps27 western blot.

**Source data 6.** *Rps27*$^{exon2del}$, *Rps27l*$^{exon2del}$ P0 brain Rps27 western blot.

**Source data 7.** *Rps27*$^{exon2del}$, *Rps27l*$^{exon2del}$ P0 liver Rps27l western blot.

**Source data 8.** *Rps27*$^{exon2del}$, *Rps27l*$^{exon2del}$ P0 heart Rps27l western blot.

**Source data 9.** *Rps27*$^{exon2del}$, *Rps27l*$^{exon2del}$ P0 forelimb Rps27l western blot.

**Source data 10.** *Rps27*$^{exon2del}$, *Rps27l*$^{exon2del}$ P0 brain Rps27l western blot.

**Source data 11.** *Rps27*$^{exon2del}$, *Rps27l*$^{exon2del}$ P0 liver Gapdh western blot.

**Source data 12.** *Rps27*$^{exon2del}$, *Rps27l*$^{exon2del}$ P0 heart Gapdh western blot.

**Source data 13.** *Rps27*$^{exon2del}$, *Rps27l*$^{exon2del}$ P0 forelimb Gapdh western blot.

**Source data 14.** *Rps27*$^{exon2del}$, *Rps27l*$^{exon2del}$ P0 brain Gapdh western blot.

**Source data 15.** Editable data from ***Figure 4B and G***.

*Rps27l*$^{exon2del / +}$ males and females (***Figure 4B***). These *Rps27l*$^{exon2del / exon2del}$ offspring died or needed to be euthanized due to animal distress by postnatal days 13–17 (P13–17). Interestingly, *Rps27*$^{exon2del / +}$ mice were also viable, but no *Rps27*$^{exon2del / exon2del}$ offspring were recovered from crosses of *Rps27*$^{exon2del / +}$ males and females (***Figure 4B***). When we dissected embryonic day 8.5 (e8.5) embryos from *Rps27*$^{exon2del / +}$ × *Rps27*$^{exon2del / +}$ crosses, only one *Rps27*$^{exon2del / exon2del}$ specimen out of 35 dissected embryos was recovered. This *Rps27*$^{exon2del / exon2del}$ embryo was severely delayed in development compared to littermates (***Figure 4C and D***). For the *Rps27* and *Rps27l* truncation allele genotypes that are viable at birth, we confirmed by western blot of multiple postnatal day 0 (P0) tissues that detectable protein expression from the respective truncated paralog is slightly diminished in *Rps27*$^{exon2del / +}$ specimens and nearly eliminated in *Rps27l*$^{exon2del / exon2del}$ specimens (***Figure 4E and F***), with residual detection due to trace antibody cross-reactivity against the intact paralog (as demonstrated in ***Figure 3—figure supplement 1A and D***). Interestingly, we observed that *Rps27l*$^{exon2del / +}$ and *Rps27l*$^{exon2del / exon2del}$ tissues had increased expression of the intact *Rps27* protein, which is analogous to previously reported instances in which depletion of an RP gene is associated with increased expression of a paralogous gene (***Milenkovic et al., 2023***; ***O'Leary et al., 2013***). Together, these findings demonstrate that impaired expression from the *Rps27* locus impacts viability at significantly earlier embryonic stages than impaired expression from the *Rps27l* locus.

The phenotypes of these *Rps27*$^{exon2del}$ and *Rps27l*$^{exon2del}$ loss-of-function alleles yield several preliminary insights on the possible roles of these RP paralogs in vivo. The fact that *Rps27* and *Rps27l* loss-of-function alleles are each lethal in homozygosity, even with both alleles of the other paralog intact, disfavors the hypothesis that *Rps27* and *Rps27l* engage in paralog buffering. The difference in the timing of lethality could be consistent with the two proteins having distinct functions; under such a model, it would appear that Rps27 protein has a critical function in very early developmental processes, while the functions of Rps27l protein are dispensable until postnatal stages. Divergent protein function could also explain why an increase in Rps27 protein is insufficient to rescue the *Rps27l*$^{exon2del / exon2del}$ genotype. However, divergent protein function is not the only possible explanation for either of these observations: the timing difference could also be explained by early reliance on expression from the *Rps27* locus, whereas this dependence later shifts to expression of an equivalent protein from the *Rps27l* locus. If the regulatory characteristics of *Rps27* and *Rps27l* are sufficiently dissimilar, it may be impossible for one paralog to compensate for the other's loss of function in specific cell

types. Therefore, with the evidence presented thus far, it remains possible that Rps27 and Rps27l are functionally identical proteins, and that the $Rps27l^{exon2del / exon2del}$ genotype is lethal because the quantity of Rps27 increase is insufficient to compensate for the Rps27l deficiency.

Previous work has shown that the early postnatal lethality of $Rps27l^{GT/GT}$ can be rescued on a $Trp53$ loss-of-function ($Trp53^{LOF/+}$ or $Trp53^{LOF/LOF}$) background. This finding was attributed to a model in which $Rps27l$ depletion impairs ribosome biogenesis and thus causes accumulation of unincorporated RPs, which increase $Trp53$ activity by blocking its degradation by Mdm2. The increased Trp53 triggers apoptosis in hematopoietic tissues (*Xiong et al., 2014*). We thus crossed the $Rps27^{exon2del}$ and $Rps27l^{exon2del}$ mice to $Trp53^{null}$ mice (*Jacks et al., 1994*). We indeed found that $Rps27l^{exon2del / exon2del};Trp53^{null / +}$ mice are viable. However, no $Rps27^{exon2del / exon2del};Trp53^{null / +}$ or $Rps27^{exon2del / exon2del};Trp53^{null / null}$ offspring were recovered from crossing $Rps27^{exon2del / +};Trp53^{null / +}$ males and females (*Figure 4G*). These results suggest that disabling expression from the $Rps27$ locus disrupts ribosome biogenesis either to a greater degree than $Rps27l$ that cannot be mitigated by $Trp53$ depletion or through a non-$Trp53$-mediated mechanism.

## Rps27 and Rps27l proteins are functionally interchangeable across all examined murine tissues

Having demonstrated that expression of both $Rps27$ and $Rps27l$ is essential to development at different stages, we next asked whether Rps27 protein can rescue loss of Rps27l protein, and vice versa, at the whole-organism level. This test is critical to understanding whether the two paralogs' gene products are functionally interchangeable, especially in light of the above finding that the $Rps27l^{exon2del / exon2del}$ genotype is early postnatal lethal despite an increased amount of Rps27 protein. In order to do so rigorously, it was ideal to express the swapped protein sequences from the endogenous genomic loci with minimal perturbation to the regulatory contexts. Using CRISPR in mouse embryos, we minimally edited the endogenous genomic loci for $Rps27$ and $Rps27l$ to encode the protein sequence of the other paralog. This yielded two novel mouse alleles: $Rps27^{Rps27l}$, which expresses Rps27l protein from the $Rps27$ locus; and $Rps27l^{Rps27}$, which expresses Rps27 protein from the $Rps27l$ locus (*Figure 5A and B*).

To assess the organism-wide impact of substituting the Rps27 protein sequence with Rps27l or vice versa, we performed a detailed characterization of $Rps27^{Rps27l}$ and $Rps27l^{Rps27}$ mice. Remarkably, heterozygous crosses for both alleles ($Rps27^{Rps27l /+} \times Rps27^{Rps27l /+}$, and $Rps27l^{Rps27 /+} \times Rps27l^{Rps27 /+}$) resulted in normal genotype frequencies among offspring of both sexes (*Figure 5C*). Western blots of adult mouse tissues confirmed loss of Rps27 protein and increased Rps27l protein in $Rps27^{Rps27l / Rps27l}$ mice (*Figure 5D and E*). This effect is most pronounced in the spleen, which contains abundant B cells, one of the cell types that highly express the $Rps27$ locus as shown above (*Figure 1D*). Likewise, loss of Rps27l protein expression with concurrently increased Rps27 protein was observed in $Rps27lRps27^{Rps27 / Rps27}$ mice (*Figure 5D and E*) and the effect is most pronounced in liver tissue, which contains hepatocytes that highly express the $Rps27l$ locus (*Figure 1D* and *Figure 1—figure supplement 2E*). This demonstrates that expressing Rps27 from the $Rps27l$ locus completely rescues the early lethality observed upon homozygous truncation of $Rps27l$, and vice versa. Later organism fitness was also rescued: pups of all genotypes gained weight at similar rates (*Figure 5F*), and male and female heterozygous and homozygous mice are viable to at least 1 year of age and are fertile. A detailed necropsy of homozygous 10–16-week-old males and age-matched wild-type controls, performed by a veterinary pathologist blinded to specimen genotype, revealed no clinically significant differences in gross organ weight, gross organ morphology, or tissue histology for either line (*Supplementary file 4*) upon examination of neurological, cardiovascular, respiratory, gastrointestinal, genitourinary, lymphatic, endocrine, hematopoietic, integumentary, and musculoskeletal tissues.

Given the cell type-specific patterns of $Rps27$ and $Rps27l$ mRNA levels that we have described above, it was important to explore whether a functional difference between the Rps27 and Rps27l proteins might only be apparent in tissues that preferentially express one paralog or the other. The top cell types of interest included mammary alveolar cells and hepatocytes, which have a high $Rps27l:Rps27$ ratio; and B, T, and NK cells, which have a low ratio (*Figure 1D*). We, therefore, devoted additional effort to characterizing the mammary gland, liver, and hematopoietic organs. A liver panel and complete blood count were performed on the 10–16-week-old male necropsy specimens to survey for anomalies in liver function or hematopoiesis. These yielded no clinically significant

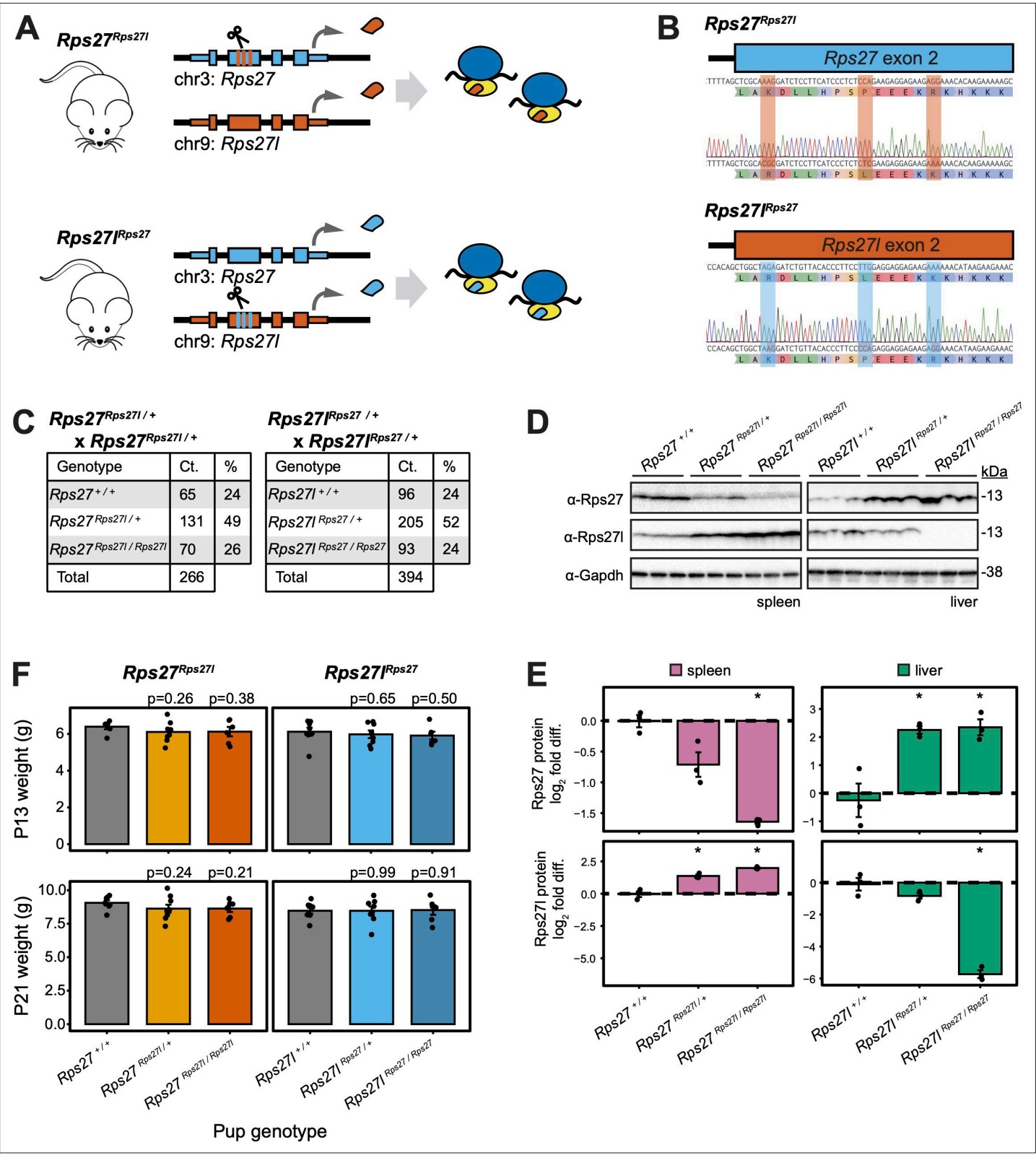

**Figure 5.** Homogenized *Rps27* and *Rps27l* alleles in mice exhibit normal genotype ratios and early development. (**A**) CRISPR editing to generate *Rps27^Rps27l^* and *Rps27l^Rps27^* homogenized mice. (**B**) Sanger sequencing of *Rps27^Rps27l^* and *Rps27l^Rps27^* homogenized mouse alleles. (**C**) Genotype ratios among live offspring of *Rps27^Rps27l / +^* and *Rps27l^Rps27 / +^* heterozygous crosses at postnatal day 13 (P13). (**D**) Western blots of adult mouse (9–10 mo) tissues with Gapdh as loading control and n = 3 biological replicates (animals) per genotype. (**E**) Rps27 and Rps27l protein abundance in (**D**) quantified as

*Figure 5 continued on next page*

*Figure 5 continued*

Gapdh-normalized $\log_2$ fold difference relative to averaged +/+ samples for each tissue. Error bars show standard error. *p<0.05 by *t*-test; unmarked = not significant. (**F**) Pup weights at P13 and P21, grouped by pup genotype. Each data point represents the average weight among pups of the indicated genotype within a litter. Only pups from the first litters born to *Rps27*<sup>Rps27l / +</sup> and *Rps27l*<sup>Rps27 / +</sup> dams are included. n = 6–9 litters per genotype. Significance versus *Rps27* <sup>+/+</sup> and *Rps27l*<sup>+/+</sup>, respectively, was assessed by *t*-test. Error bars show standard error. See *Figure 5—source data 1–7*; see also *Figure 5—figure supplement 1*.

The online version of this article includes the following source data and figure supplement(s) for figure 5:

**Source data 1.** *Rps27*<sup>Rps27l</sup> adult spleen Rps27 western blot.

**Source data 2.** *Rps27*<sup>Rps27l</sup> adult spleen Rps27l western blot.

**Source data 3.** *Rps27*<sup>Rps27l</sup> adult spleen Gapdh western blot.

**Source data 4.** *Rps27l*<sup>Rps27l</sup> adult liver Rps27 western blot.

**Source data 5.** *Rps27l*<sup>Rps27l</sup> adult liver Rps27l western blot.

**Source data 6.** *Rps27l*<sup>Rps27l</sup> adult liver Gapdh western blot.

**Source data 7.** Editable data from *Figure 5C*.

**Figure supplement 1.** Comparison of litter size (number of pups) and days between start of mating and first birth for all dam genotypes of homogenized *Rps27*<sup>Rps27l</sup> and *Rps27l*<sup>Rps27</sup> lineages.

---

differences between genotypes (*Figure 6A*, *Supplementary file 5*). To assess the mammary gland, we focused on the pregnancy and lactation stages because precursors to alveolar cells emerge and mature during these stages (*Macias and Hinck, 2012*), and because we had detected fluctuations in *Rps27* and *Rps27l* mRNA abundance in bulk tissues from these timepoints relative to nulliparous samples (*Figure 1E and F* and *Figure 1—figure supplement 2D and E*). We first tracked pup weight gain as a metric of mammary gland function. Nulliparous littermate-matched females at 8 weeks of age were housed with stud males and separated from the males before parturition. Since genetically manipulated mouse lines can exhibit lactation failure at a range of timepoints (*Palmer et al., 2006*), litter size and pup weight were assessed at P4, P13, and P21 (*Figure 6B*). No statistically significant differences in pup weight gain were detected between any of the maternal genotypes. There were also minimal differences in litter size or maternal age at parturition (*Figure 5—figure supplement 1*).

We considered the possibility that female mice housed under standard laboratory conditions may have a substantial reserve of lactation capacity, which could mask a moderate effect of *Rps27(l)* homogenization on mammary gland development and function. We, therefore, used carmine alum staining and hematoxylin-eosin staining to assess the gross morphology and mammary fat pad filling of nulliparous, pregnant, and lactating females (*Figure 6C*). No anomalies in mammary gland morphogenesis were detected, with all genotypes exhibiting similar epithelial branch length and number, fat pad filling, alveolar size, and alveolar wall thickness. From this evidence, we conclude that homogenization of either the *Rps27* or *Rps27l* locus has no effect on overall mouse fitness and also no effect on the morphology or function of tissues that preferentially express either *Rps27* or *Rps27l*. These findings suggest that the Rps27 and Rps27l proteins are functionally similar in the setting of normal organism physiology, even in tissues that preferentially express one paralog.

### *Rps27* and *Rps27l* homogenization do not affect health in later life or in response to genotoxic stress

Having assessed for phenotypes in young homogenized male and female mice, we considered the possibility that the Rps27 and Rps27l proteins may function similarly under optimal conditions of homeostasis in young animals, but could act differently to confer an evolutionary benefit at a later age or under stress. To assess this hypothesis, we co-housed *Rps27*<sup>Rps27l</sup> and *Rps27l*<sup>Rps27</sup> homozygous, heterozygous, and wild-type male littermates under standard husbandry conditions (see 'Materials and methods') until 9–10 months of age. At that time, we weighed the mice and performed a complete blood count and liver panel, again targeting the organ systems with preferential *Rps27* or *Rps27l* expression. No clinically significant differences were observed between genotypes in any of the included assays (*Figure 7A*, *Supplementary file 6*). Thus, homogenization of *Rps27* or *Rps27l* has no detectable effect on the physiology of these organs, even later in life. These findings diminish the

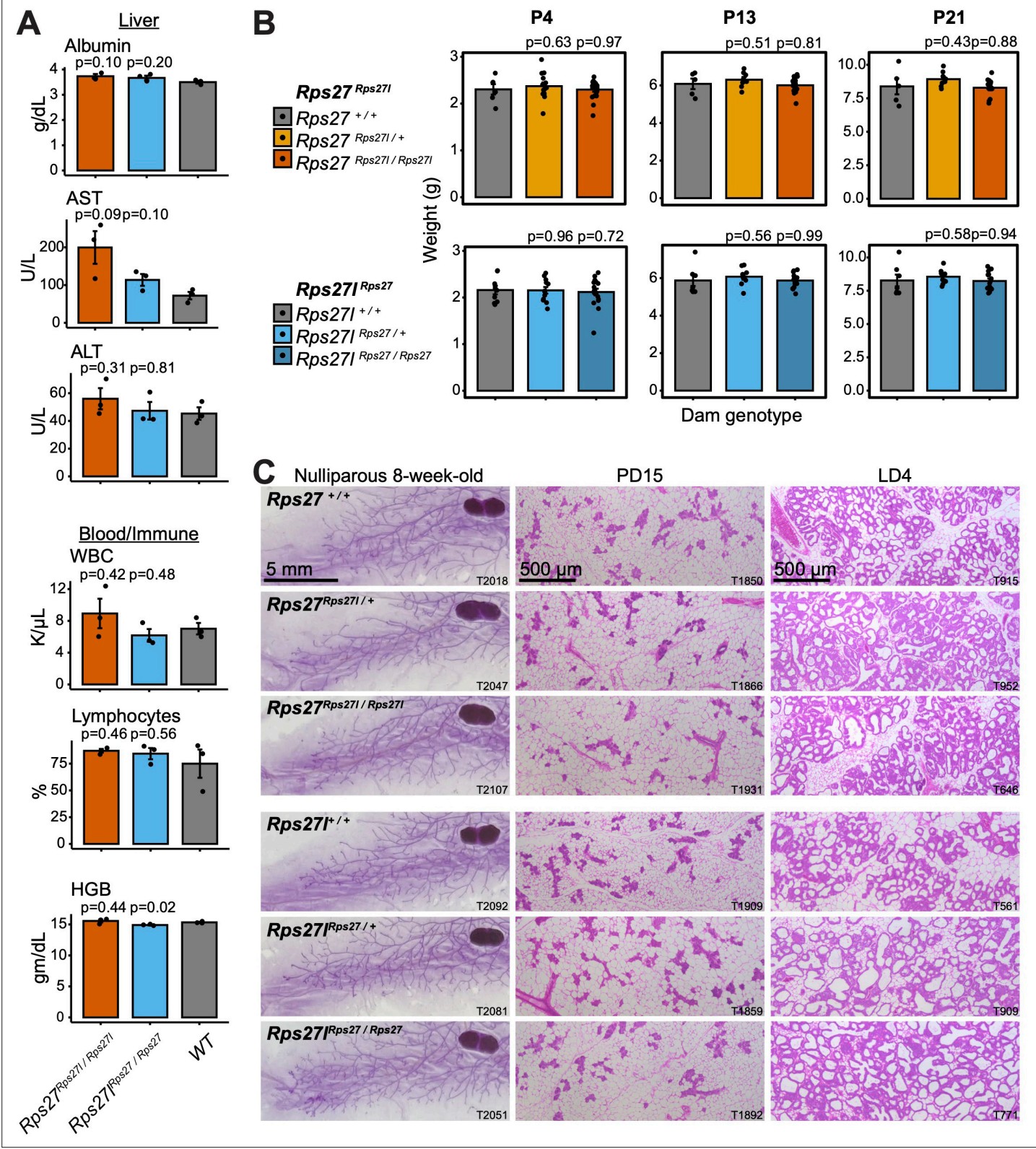

**Figure 6.** Homogenization of *Rps27* and *Rps27l* does not impact tissues that preferentially express one paralog. (**A**) Selected biomarkers from bloodwork performed on 10–16-week-old homozygous males and age-matched wild-type controls from the *Rps27^Rps27l^* and *Rps27l^Rps27^* mouse lines. See **Supplementary file 5** for additional biomarkers. AST, aspartate aminotransferase; ALT, alanine aminotransferase; WBC, white blood cells; HGB, hemoglobin. n = 3 biological replicates (individual animals) per genotype. Significance relative to WT was assessed by *t*-test. Error bars show standard

*Figure 6 continued on next page*

*Figure 6 continued*

error. (**B**) Mean pup weight per litter at postnatal days (PDs) 4, 13, and 21, grouped by dam genotype. Only pups from a dam's first litter are included. n = 5–19 litters per dam genotype and timepoint. Significance relative to WT was assessed by *t*-test. Error bars show standard error. (**C**) Representative images of carmine alum-stained or hematoxylin-eosin (H&E)-stained mammary glands at the indicated stages. T, animal ear tag number. See also *Figure 6—source data 1–18*.

The online version of this article includes the following source data for figure 6:

**Source data 1.** T2018; *Rps27*$^{+/+}$ nulliparous 8-week-old.

**Source data 2.** T2047; *Rps27*$^{Rps27l\,/+}$ nulliparous 8-week-old.

**Source data 3.** T2107; *Rps27*$^{Rps27l\,/\,Rps27l}$ nulliparous 8-week-old.

**Source data 4.** T2092; *Rps27l*$^{+/+}$ nulliparous 8-week-old.

**Source data 5.** T2081; *Rps27l*$^{Rps27\,/+}$ nulliparous 8-week-old.

**Source data 6.** T2051; *Rps27lRps27*$^{Rps27\,/\,Rps27}$ nulliparous 8-week-old.

**Source data 7.** 5x_t1850_1; *Rps27*$^{+/+}$ pregnancy day 15.

**Source data 8.** 5x_t1866_1; *Rps27*$^{Rps27l\,/+}$ pregnancy day 15.

**Source data 9.** 5x_t1931_1; *Rps27*$^{Rps27l\,/\,Rps27l}$ pregnancy day 15.

**Source data 10.** 5x_t1909_; *Rps27l*$^{+/+}$ pregnancy day 15.

**Source data 11.** 5x_t1859_1; *Rps27l*$^{Rps27\,/+}$ pregnancy day 15.

**Source data 12.** 5x_t1892_1; *Rps27lRps27*$^{Rps27\,/\,Rps27}$ pregnancy day 15.

**Source data 13.** 5x_t915_02; *Rps27*$^{+/+}$ lactation day 4.

**Source data 14.** 5x_t952_2; *Rps27*$^{Rps27l\,/+}$ lactation day 4.

**Source data 15.** 5x_t646_3; *Rps27*$^{Rps27l\,/\,Rps27l}$ lactation day 4.

**Source data 16.** 5x_t561_02; *Rps27l*$^{+/+}$ lactation day 4.

**Source data 17.** 5x_t909_; *Rps27l*$^{Rps27\,/+}$ lactation day 4.

**Source data 18.** 5x_t771_03; *Rps27lRps27*$^{Rps27\,/\,Rps27}$ lactation day 4.

likelihood that there might be functional differences between the *Rps27* and *Rps27l* protein sequences that have cumulative effects over the mouse lifespan.

Lastly, we tested the effects of stress stimuli on ex vivo cells derived from *Rps27*$^{Rps27l}$ and *Rps27l*$^{Rps27}$ mice. It has previously been reported that *Trp53* differentially regulates *Rps27* and *Rps27l* expression (*He and Sun, 2007*; *Li et al., 2007*; *Xiong et al., 2011*). Furthermore, Rps27 and Rps27l proteins reportedly bind and regulate the *Trp53*-regulating ubiquitin ligase Mdm2 with different affinity in vitro, and thus may form distinct feedback loops impacting *Trp53* activity (*Xiong et al., 2011*). Even though the expression patterns of *Rps27* and *Rps27l* that we observed are not likely driven by *Trp53* (*Figure 2A and B*), we hypothesized that functional differences between the Rps27 and Rps27 proteins might be revealed under the types of genotoxic or cell cycle-related stress conditions for which *Trp53* is classically a master regulator of response. We isolated mouse embryonic fibroblasts (MEFs) from homozygous, heterozygous, and wild-type mice of the *Rps27*$^{Rps27l}$ and *Rps27l*$^{Rps27}$ lineages. We first analyzed their distribution across cell cycle phases when cultured in vitro and found that MEFs of all genotypes had similar frequency in each phase (*Figure 7B*). We then treated them with varying doses of doxorubicin and etoposide, two chemotherapeutic drugs that induce DNA damage and consequently activate *Trp53*. The number of viable and metabolically active cells was assessed after 24–48 hr of drug treatment (*Figure 7C*). Higher doses of doxorubicin and etoposide consistently resulted in decreased cell viability, yet the same degree of effect was seen across all genotypes. Thus, for the purposes of cellular survival and proliferation under doxorubicin or etoposide treatment, the Rps27 and Rps27l proteins also appear to be interchangeable.

## Discussion

In this work, we set out to test two categories of hypotheses regarding the evolutionary retention of a mammalian RP paralog pair: either that the paralogs encode functionally distinct proteins or that their essentiality is due to dosage sharing or paralog buffering. We observed that *Rps27* and *Rps27l* mRNA abundance is inversely correlated and cell type-dependent across healthy mouse tissues, showed

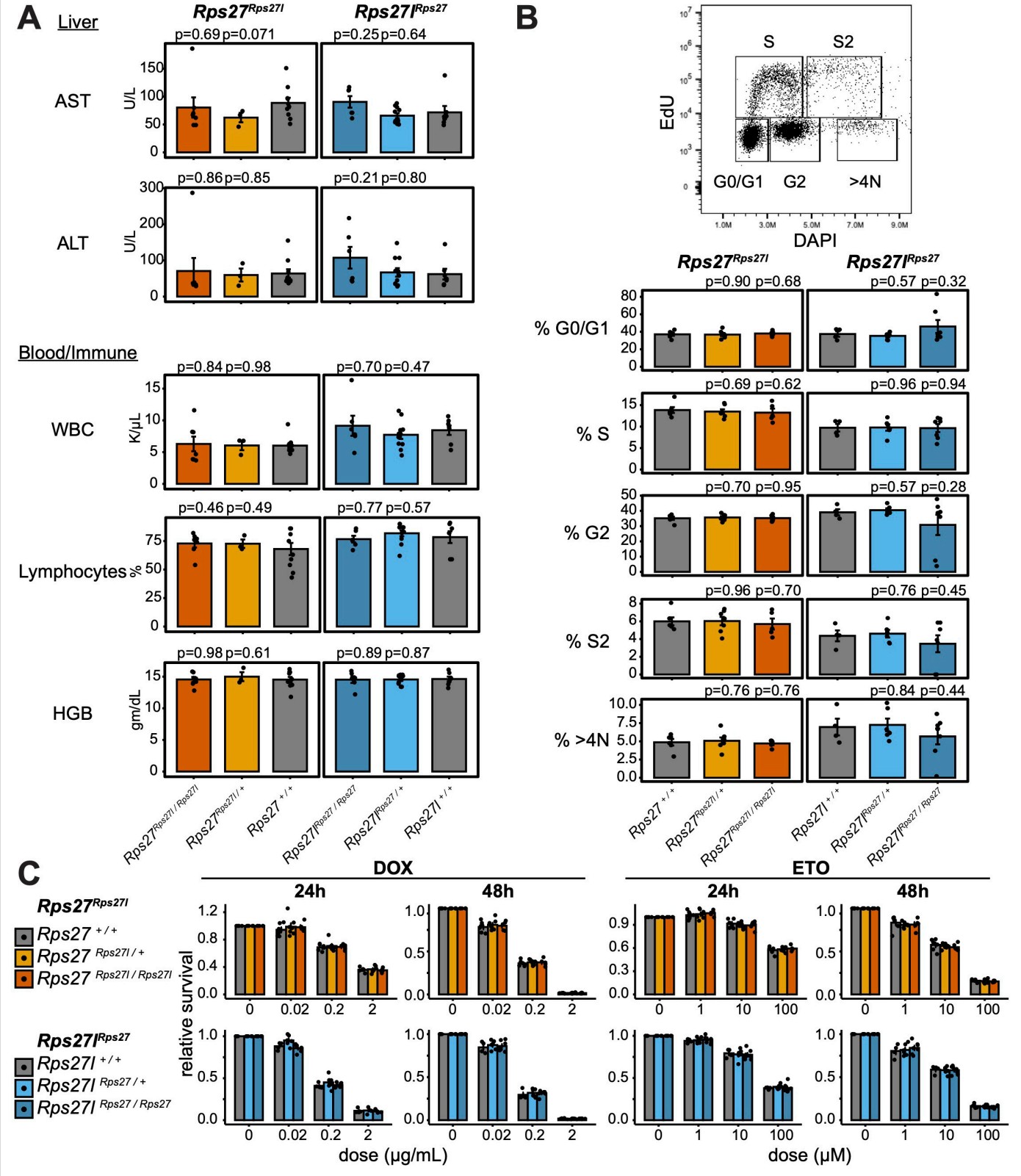

**Figure 7.** *Rps27* and *Rps27l* homogenization does not affect physiology at later age or impact response to genotoxic stress. (**A**) Biomarkers from 9- to 10-month-old *Rps27^Rps27l^* and *Rps27l^Rps27^* homozygous and heterozygous males and age-matched wild-type (WT) controls. n = 3–11 animals per genotype. See **Supplementary file 6** for additional biomarkers. AST, aspartate aminotransferase; ALT, alanine aminotransferase; WBC, white blood cells; HGB, hemoglobin. (**B**) Percent of singlet mouse embryonic fibroblasts (MEFs) in each cell cycle phase as measured by EdU/DAPI flow cytometry. n

*Figure 7 continued on next page*

*Figure 7 continued*

= 4–7 biological replicates (MEF lines isolated from single embryos). (**C**) Fraction of viable MEFs relative to untreated controls after doxorubicin (DOX) or etoposide (ETO) treatment. n = 4–7 biological replicates; n = 2 technical replicates (separate wells) per biological replicate. For all panels, significance relative to WT was assessed by *t*-test and error bars show standard error. For (**C**), no comparisons were significant at p<0.05.

that Rps27- and Rps27l-ribosomes differentially associate with transcripts of cell cycle-related genes, and demonstrated that loss-of-function alleles of *Rps27* and *Rps27l* are both homozygous lethal but manifest at different developmental stages. Based on these findings, divergent protein functions and dosage sharing through subfunctionalized expression were both plausible reasons for *Rps27* and *Rps27l* conservation, while paralog buffering was not. Ultimately, after extensive examination of *Rps27^{Rps27l}* and *Rps27l^{Rps27}* mice, we concluded that the Rps27 and Rps27l proteins are functionally interchangeable. Together, these results suggest that *Rps27* and *Rps27l*, which most likely arose alongside other RP gene duplicates during a whole-genome duplication, have been evolutionarily retained because the divergence of their expression patterns has resulted in both genes becoming necessary for achieving adequate total expression of the RP across cell types (*Figure 8*).

Several questions arise when considering the in vivo homogenized mouse outcomes alongside the molecular findings reported here and in previous literature. We detected preferential association of Rps27- or Rps27l-ribosomes with cell cycle-related transcripts, yet replacing Rps27 protein with Rps27l and vice versa had no detectable impact on cell cycle progression (*Figure 7B*). One explanation is that Rps27- and Rps27l-ribosomes may have cell cycle-dependent abundance and therefore differentially encounter cyclically expressed transcripts, a consideration that remains to be addressed in future work. Another possibility is that Rps27 and Rps27l proteins do affect ribosome affinity for specific transcripts, but other pathways compensate to maintain a normal cell cycle in homogenized cells.

From previous literature, some evidence may suggest that Rps27 and Rps27l do have distinct protein functions, especially in *Trp53*-related signaling pathways and apoptotic processes resulting from genotoxic stimuli (*He and Sun, 2007*; *Li et al., 2007*; *Xiong et al., 2020*; *Xiong et al., 2018*; *Zhao et al., 2018*). Depletion or overexpression of *Rps27l* was reported to impact *Trp53* activity differently than *Rps27* depletion or overexpression (*Xiong et al., 2014*; *Xiong et al., 2011*). Purified Rps27 and Rps27l protein bind the *Trp53* ubiquitin ligase Mdm2 in vitro with different affinity and are degraded by Mdm2 at different rates, which depends on the N-terminal portion of Rps27 and Rps27l where the differing residues reside (*Xiong et al., 2011*). Also, knockdown of *Rps27l* but not *Rps27* was reported to reduce the levels of DNA repair proteins FANCD2 and FANCI (*Sun et al., 2020*). However, in our comparison of WT and homogenized cells under genotoxic stimuli known to activate the *Trp53-Mdm2* axis (*Figure 7C*), we did not detect any differences in cell survival that would suggest functional divergence between the Rps27 and Rps27l proteins. We acknowledge the possibility that the two proteins could have distinct functions under environmental conditions not tested here or that the effects of homogenization are masked by compensation in other cellular pathways. Nevertheless, the normal physiology of both homogenized mouse lines until at least 9–10 months of age suggests that total expression of *Rps27* and *Rps27l* impacts in vivo fitness more so than differences in protein characteristics. Importantly, our findings regarding homogenization involved no knockdown, knockout, or overexpression of RPs, thereby minimizing any indirect effects of perturbing RP expression.

The conclusion that *Rps27* and *Rps27l* likely persisted across evolution due to dosage sharing emphasizes the principle that distinct protein sequences and distinct expression patterns do not always indicate a paralog with tissue-specific protein functions. With the greater ease of genetic editing afforded by CRISPR, it would be fascinating to apply the endogenous epitope tagging and homogenization approaches used here to other mammalian RP paralogs that also have tissue- or cell type-dependent expression. *Rpl3l* (*uL3L*), for example, is an RP paralog that only expresses in skeletal and cardiac myocytes, where *Rpl3* (*uL3*) is not expressed (*Guimaraes and Zavolan, 2016*). While knockdowns and knockouts of *Rpl3l* in cells and mice have been characterized (*Chaillou et al., 2016*; *Kao et al., 2021*; *Milenkovic et al., 2021*), a homogenized *Rpl3l^{Rpl3}* mouse allele would be invaluable for determining whether Rpl3l-ribosomes have myocyte-specific functions. As another example, *Rpl10l* (*uL16L*) and *Rpl39l* (*eL39L*) are mainly expressed in the testes and likely compensate during spermatogenesis for their respective progenitor genes, *Rpl10* (*uL16*) and *Rpl39* (*eL39*); the latter are encoded on the X chromosome and are therefore not expressed after meiotic X chromosome inactivation (*Guimaraes*

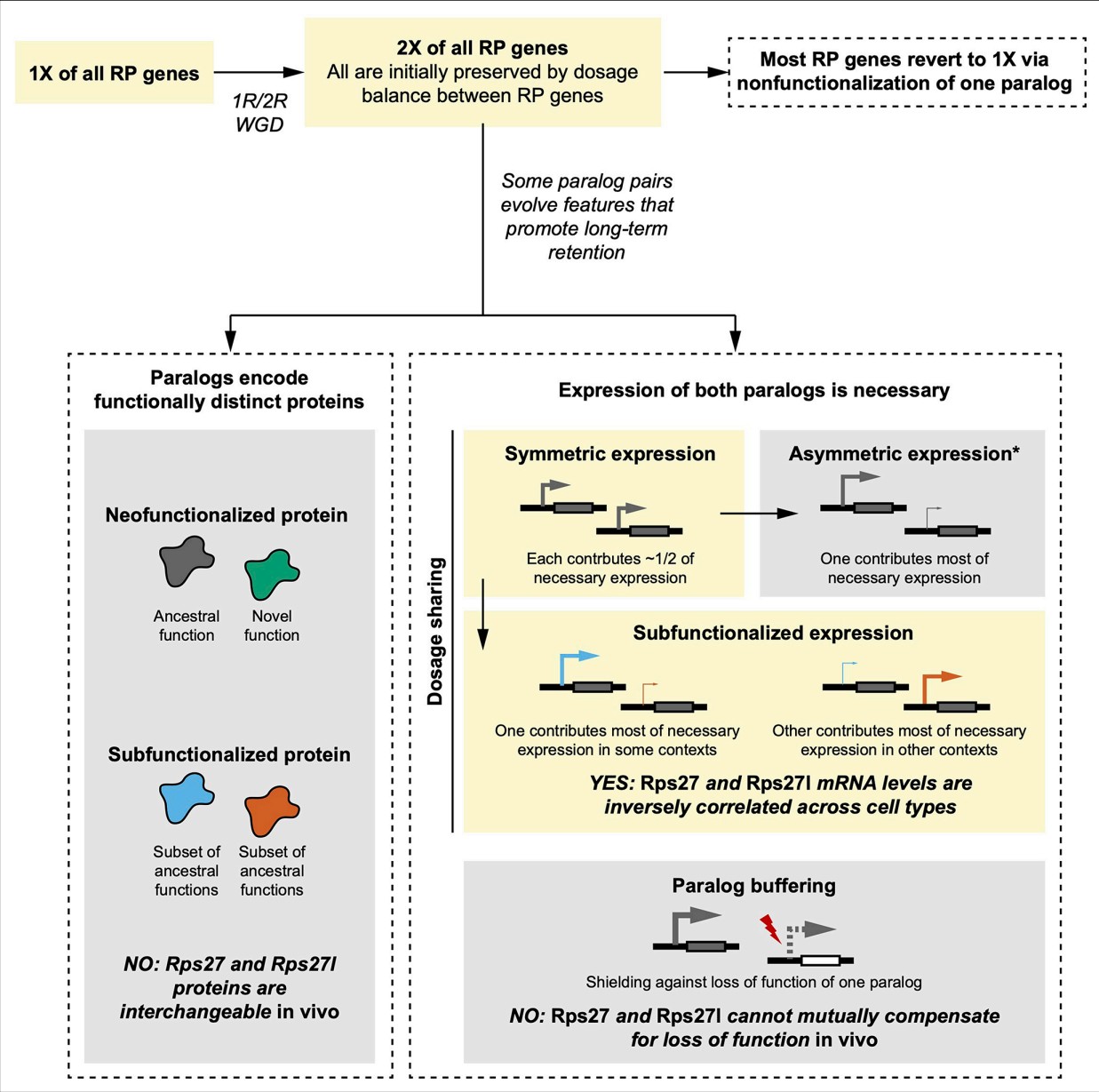

**Figure 8.** Comparison of empirical *Rps27* and *Rps27l* characteristics with hypothesized evolutionary trajectories. Yellow boxes indicate the most likely trajectory for *Rps27* and *Rps27l*. In brief, a whole-genome duplication (WGD) during early vertebrate evolution duplicated all RP genes. This mechanism is probable because the maintained dosage balance between RPs would have promoted initial preservation of duplicates. While most RP genes then reverted to single genes via nonfunctionalization of one paralog, some paralog pairs evolved features that promoted retention. Immediately after duplication, *Rps27* and *Rps27l* probably had similar regulatory elements and thus would have expressed symmetrically, but our findings suggest that *Rps27* and *Rps27l* now exhibit subfunctionalized expression that renders both paralogs necessary to achieve the requisite total expression of this RP. We found no evidence of functional differences between *Rps27* and *Rps27l* protein, nor successful compensation by either paralog for a loss-of-function of the other. Not pictured here are neofunctionalized expression and beneficial dosage increase; while these modes of paralog retention have been observed for other genes, they are less relevant for RP paralogs if it is assumed that excess dosage of an individual RP gene is not advantageous. *Symmetric expression frequently shifts towards asymmetric expression, which can be an intermediate state towards nonfunctionalization of the minor paralog.

*and Zavolan, 2016*; *Sugihara et al., 2010*). Knockouts of each have been characterized in vivo and in vitro, respectively: exogenous expression of *Rpl10* partially rescues spermatogenesis in the *Rpl10L* knockout (*Jiang et al., 2017*), whereas *Rpl39L* knockout in vivo also impairs spermatogenesis, and in vitro Rpl39L knockout with attempted rescue by *Rpl39* overexpression suggests that the *Rpl39* paralogs are non-interchangeable with respect to functions related to nascent peptide folding (*Li*

*et al., 2022*; *Zou and Qi, 2021*; *Zou et al., 2021*). In both cases, a homogenization approach would aid in determining whether the residual functional defect is because the rescue does not precisely recapitulate the endogenous expression pattern of the deleted paralog or because the overexpressed paralog lacks some specific function performed by the depleted protein. It is important to note that the properties of *Rps27* and *Rps27l* observed here do not necessarily extend to all mammalian RP paralogs, and that some mammalian RP paralog pairs, such as *Rpl3/Rpl3l* and *Rpl22/Rpl22l* (*eL22/eL22L*), have more differences in amino acid sequence than *Rps27/Rps27l*. Our work reinforces the importance of directly comparing protein function between RP paralogs in an endogenous context.

Lastly, our findings raise questions about the two *Rps27* copies that have long been asked about gene duplication in general: Did the ancestral *Rps27* have some characteristic that led to its duplication and preservation? Is the existence of two copies with divergent expression but equivalent proteins evolutionarily beneficial? Or did their preservation and cell type-dependent expression patterns result from genetic drift without adaptive selection? It should first be noted that gene duplication is common (*Lynch and Conery, 2000*), and that few duplicates persist while most degrade. It should also be noted that not all gene features arising during evolution are necessarily beneficial, and that neutral evolution may play a large role in determining which duplicates persist and how their regulatory elements evolve (*Lynch, 2007*). RPs, in general, may have a propensity for paralog retention: in *Paramecium* and yeast, for example, duplicates of highly translated genes or components of protein complexes often persist, including ribosome, histone, or cytoskeleton proteins (*Aury et al., 2006*; *Wapinski et al., 2007*). Among present-day mammalian RPs, however, existing as a paralog pair puts *Rps27* in the minority. A final curious observation that we (*Figure 1B*) and others have made is that, in teleost fish species that underwent 3R and 4R WGD, more than four *Rps27* copies have been reported, whereas most other RPs have reverted to fewer copies (*Kuang et al., 2020*; *Manchado et al., 2007*). Is there something about *Rps27* that causes its paralogs to persist when other RP paralogs generated by WGD do not? Perhaps it is advantageous to have multiple copies whose expression could be more finely regulated (*Greer et al., 2000*) or perhaps the ancestral Rps27 protein was already pleiotropic in its expression or protein function and was thus amenable to divergence into two genes with different expression and possibly subtle protein differences (*Conant and Wolfe, 2008*; *Prince and Pickett, 2002*). More work is needed to further probe the evolution and function of mammalian RP paralogs, and to provide additional comparisons between theoretical and experimental perspectives on paralog evolution. As a paradigm, the studies we report here set the groundwork for future investigation of RP paralog function.

# Materials and methods

**Key resources table**

| Reagent type (species) or resource | Designation | Source or reference | Identifiers | Additional information |
|---|---|---|---|---|
| Gene (*Mus musculus*) | *Rps27* (*eS27*) | Ensembl version 109 | ENSMUSG00000090733 | |
| Gene (*M. musculus*) | *Rps27l* (*eS27L*) | Ensembl version 109 | ENSMUSG00000036781 | |
| Gene (*Homo sapiens*) | *RPS27* (*eS27*) | Ensembl version 109 | ENSG00000177954 | |
| Gene (*H. sapiens*) | *RPS27L* (*eS27L*) | Ensembl version 109 | ENSG00000185088 | |
| Gene (multiple species) | *Rps27* and *Rps27l* orthologs | Multiple | | See *Figure 1*, *Supplementary file 1* for full details |
| Strain, strain background (*M. musculus*) | *Rps27exon2del*, C57BL/6J | This paper | | See 'Generation of genetically edited mouse lines,' *Figure 4*, *Supplementary file 7* |
| Strain, strain background (*M. musculus*) | *Rps27lexon2del*, C57BL/6J | This paper | | See 'Generation of genetically edited mouse lines,' *Figure 4*, *Supplementary file 7* |
| Strain, strain background (*M. musculus*) | *Rps27Rps27l*, C57BL/6J | This paper | | See 'Generation of genetically edited mouse lines,' *Figure 4*, *Supplementary file 7* |

*Continued on next page*

*Continued*

| Reagent type (species) or resource | Designation | Source or reference | Identifiers | Additional information |
|---|---|---|---|---|
| Strain, strain background (*M. musculus*) | *Rps27l^{Rps27}*, C57BL/6J | This paper | | See 'Generation of genetically edited mouse lines,' *Figure 4*, *Supplementary file 7* |
| Strain, strain background (*M. musculus*) | *Trp53^{null}*, C57BL/6J | Jackson Laboratories https://doi.org/10.1016/s0960-9822(00)00002-6 | Strain #002101; RRID:IMSR_JAX:002101 | |
| Cell line (*M. musculus*) | *Rps27-FLAG* mESCs | This paper | | See 'Generation of CRISPR-edited mESCs,' *Figure 3*, *Supplementary file 7* |
| Cell line (*M. musculus*) | *Rps27l-FLAG* mESCs | This paper | | See 'Generation of CRISPR-edited mESCs,' *Figure 3*, *Supplementary file 7* |
| Antibody | Rps27 (goat polyclonal) | Thermo Fisher | PA5-18092; RRID:AB_10980328 | 1:10,000 |
| Antibody | Rps27l (rabbit polyclonal) | ProteinTech | 15871-1-AP; RRID:AB_2253903 | 1:1000 |
| Antibody | Actb (mouse monoclonal) | Cell Signaling | 3700S; RRID:AB_2242334 | 1:1000 |
| Antibody | Rps3 (mouse monoclonal) | Abcam | ab77330; RRID:AB_1566697 | 1:1000 |
| Antibody | FLAG (mouse monoclonal) | MilliporeSigma | F3165; RRID:AB_259529 | 1:1000 |
| Antibody | Gapdh (mouse monoclonal) | Thermo Fisher | AM4300; RRID:AB_2536381 | 1:5000 |
| Antibody | Trp53 (rabbit polyclonal) | Leica Biosystems | CM5; RRID:AB_563933 | 1:1000 |
| Sequence-based reagent | qPCR primers | This paper | | See *Supplementary file 7* |
| Sequence-based reagent | Genotyping primers | This paper | | See *Supplementary file 7* |
| Sequence-based reagent | Ribosome profiling primers | https://doi.org/10.1016/j.ymeth.2017.05.028 | | See also *Supplementary file 7* |
| Commercial assay or kit | NucleoSpin RNA | Macherey-Nagel | 740955.50 | |
| Commercial assay or kit | PureLink RNA Mini | Thermo Fisher | 12183018A | |
| Commercial assay or kit | ProteoExtract protein precipitation | Calbiochem | 539180 | |
| Commercial assay or kit | Direct-Zol Microprep Kit | Zymo | R2060 | |
| Commercial assay or kit | Zymo Oligo Clean & Concentrator | Zymo | D4060 | |
| Commercial assay or kit | RiboZero Gold | Illumina | 20020598 | |
| Commercial assay or kit | Zymo DNA Clean & Concentrator | Zymo | D4003 | |
| Commercial assay or kit | EdU labelling kit | Click Chemistry Tools | 1381 | |
| Software, algorithm | MEGA11 | https://doi.org/10.1093/molbev/msab120 | | |
| Software, algorithm | RAxML-ng | https://doi.org/10.1093/bioinformatics/btz305 | | |
| Software, algorithm | Cutadapt version 2.4 | https://doi.org/10.14806/ej.17.1.200 | | |
| Software, algorithm | fastx_barcode_splitter.pl | http://hannonlab.cshl.edu/fastx_toolkit/ | | |
| Software, algorithm | umi_tools version 1.0.1 | https://doi.org/10.1101/gr.209601.116 | | |
| Software, algorithm | fastq_quality_filter | http://hannonlab.cshl.edu/fastx_toolkit/ | | |
| Software, algorithm | bowtie2 version 2.3.4.3 | https://doi.org/10.1038/nmeth.1923 | | |

*Continued*

| Reagent type (species) or resource | Designation | Source or reference | Identifiers | Additional information |
|---|---|---|---|---|
| Software, algorithm | EdgeR | https://doi.org/10.1093/bioinformatics/btp616 | | |
| Software, algorithm | Voom | https://doi.org/10.1186/gb-2014-15-2-r29 | | |

## Phylogenetic analysis

To detect *Rps27* orthologs in representative animal species, NCBI BLASTP was performed through the Ensembl and RefSeq interfaces against the annotated proteins associated with the reference genome assemblies listed in *Supplementary file 1*, with human RPS27 protein (ENSP00000499044) as query and default BLASTP parameters. Search hits were excluded if their corresponding transcripts contained a single coding exons (indicating likely processed pseudogene) or if they overlapped a higher-scoring protein at the same genomic locus. The majority of protein annotations used in the analysis are Ensembl or RefSeq gene models; exceptions are noted in *Supplementary file 1*. tBLASTn was also performed to detect unannotated paralogous protein-coding genes and corroborated by RNA-Seq-based gene models when available. Protein sequences and coding sequences were downloaded from Ensembl and RefSeq. Several orthologs were annotated to have N-terminal extensions relative to the human RPS27 proteins sequence; for calculation of similarity, these were truncated.

For multispecies sequence alignment and phylogenetic tree construction, protein and coding sequence alignment was performed in the MEGA11 interface using the MUSCLE algorithm and default parameters. The codon-aware setting was used for coding sequences. Phylogenetic tree construction was performed with RAxML-ng in the RAxML graphical user interface (*Kozlov et al., 2019*). Parameters were optimized using the ModelTest-NG module (v0.1.7). Tree construction was performed with 50 starting trees and 100 bootstrap replicates.

## Animal husbandry

All animal work was reviewed and approved by the Stanford Administrative Panel on Laboratory Animal Care (APLAC, protocol #27463). The Stanford APLAC is accredited by the American Association for the Accreditation of Laboratory Animal Care. All mice used in the study were housed at Stanford University except where otherwise noted. CRISPR-edited mouse lines were generated at the Gladstone Institute Transgenic Gene Targeting Core (San Francisco, CA). All animal procedures were approved by the Institutional Animal Care and Use Committee at the University of California, San Francisco (protocol #AN180952-01B). Mouse lines were maintained on a C57BL6/J background unless otherwise stated. *Trp53^null* (*Jacks et al., 1994*) mice were purchased from Jackson Laboratories (strain #002101, Bar Harbor, ME). Mice were housed under 12 hr light–dark cycles with ad libitum irradiated chow (Teklad 2018SX, Envigo, Madison, WI), acidified water, and filtered air flow. For timed pregnancies, 1–2 female mice were housed overnight with one adult male and examined daily for vaginal plugs. Embryo stage was considered to be E0.5 on the day the vaginal plug was observed. Mice used in the same experiment were colony-matched, and also littermate-matched whenever possible. Adult mice were euthanized by $CO_2$ inhalation and confirmed by cervical dislocation per APLAC guidelines. Neonatal mice were euthanized by decapitation per APLAC guidelines. Genotyping was performed using standard PCR protocols for MyTaq HotStart Red Mix (Bioline BIO-25048, Memphis, TN) with primers listed in *Supplementary file 7*.

## Reanalysis of RNA-seq datasets

Analysis scripts are available on GitHub (see 'Data availability statement'). Gene count matrices and cell or sample type annotations were downloaded from the Mouse Cell Atlas (*Han et al., 2018*), Tabula Muris (*Tabula Muris Consortium, 2018*), *Bach et al., 2017*, and *Fu et al., 2015*. For scRNA-seq datasets, read counts from single cells were pooled. Total reads of each RP gene in a cell type were normalized by the total read count of all RP genes in the cell type to normalize for the cell type-specific rate of ribosome production, then multiplied by 100. For non-RP genes, normalized read counts are reported as reads per million.

## RT-qPCR from mouse mammary glands, heart, liver, and brain

For *Figure 2A*, whole mammary gland tissue was harvested from abdominal glands at the indicated timepoints. The central lymph node was removed, and total RNA was isolated using NucleoSpin RNA (Macherey-Nagel 740955.50, Düren, Germany) according to the manufacturer's instructions. For *Figures 1F and 2E*, tissues were harvested, cut into ~3 mm pieces, and snap-frozen in liquid nitrogen. For mammary glands, the central lymph node was removed prior to freezing. Snap-frozen tissues were powderized in a ceramic mortar and pestle while submerged in liquid nitrogen. Powder was suspended in TRIzol (Thermo Fisher 15596-018, Waltham, MA). RNA was extracted according to the manufacturer's protocol and isolated using the PureLink RNA Mini kit (Thermo Fisher 12183018A). Samples were treated with Turbo DNAse and inactivated (Thermo Fisher AM1907).

For all samples, 1 µg of RNA was reverse-transcribed using iScript RT Supermix (Bio-Rad 1708841, Hercules, CA). qPCR was performed using SsoAdvanced SYBR Green Super Mix (Bio-Rad 1725270) on a Bio-Rad CFX384 using the primers listed in *Supplementary file 7*. Two technical replicates were performed for each of the three biological replicates per condition. Ct values were normalized to a housekeeping gene or another RP gene as stated in figure legends and displayed as a fold difference relative to a reference sample.

## Mouse embryonic stem cell (mESC) culture

E14Tg2a.4 mESCs (*Smith and Hooper, 1987*) were a gift from Thom Saunder's lab (University of Michigan). Cell line identity was verified via short tandem repeat profiling performed by the American Type Culture Collection (ATCC, 137-XV, Manassas, VA). Cells were cultured in Knockout DMEM (Thermo Fisher 10829-018) supplemented to a final concentration of 15% ES-qualified fetal bovine serum (MilliporeSigma ES-009-B, Burlington, MA), 1% non-essential amino acids (MilliporeSigma TMS-001-C), 2 mM L-glutamine (MilliporeSigma TMS-002-C), 1% penicillin/streptomycin (Thermo Fisher 15140-122), 55 µM beta-mercaptoethanol (Thermo Fisher 21985-023), and 1000 U/mL mouse leukemia inhibitory factor (mLIF, Gemini 400-495 $10^7$, West Sacramento, CA). Media was changed daily and cells were passaged every 2 d. To passage, plates with ~70% confluent colonies of mESCs were washed in Dulbecco's phosphate buffered saline (DPBS) (Thermo Fisher 14190-250) and trypsinized (0.05% trypsin, dilution of Thermo Fisher 15400-054 in DPBS) for 5 min at 37°C. Trypsin was neutralized by a double volume of media. Cells were immediately dissociated by vigorous pipetting, pelleted at 200 × *g* for 3 min at room temperature, and resuspended in fresh warmed media for plating. Fresh plates were pre-coated at 37°C overnight with 0.1% gelatin (MilliporeSigma ES-006-B), which was aspirated prior to plating cells. Unless otherwise stated, cells were plated at a density equivalent to $5 \times 10^6$ cells per 10 cm plate. Mycoplasma testing was performed according to the manufacturer's instructions using the PromoKine PCR Mycoplasma Test Kit (PK-CA91-1096, Heidelberg, Germany).

## Density gradient fractionation

Gradient lysis buffer: 20 mM Tris pH 7.5 (Thermo Fisher AM9850G, AM9855G), 150 mM NaCl (Thermo Fisher AM9760G), 15 mM $MgCl_2$ (Thermo Fisher AM9530G), 1% v/v Triton-X 100 (MilliporeSigma X100-500ML), 8% v/v glycerol (MilliporeSigma G6279), 1 mM DTT (MilliporeSigma 43815), 1X cOmplete mini protease inhibitor EDTA-free (Roche 11836170001, Basel, Switzerland), 0.5% w/v deoxycholate (MilliporeSigma S1827), 100 µg/mL cycloheximide (MilliporeSigma C7698), 0.02 U/µL Turbo DNAse (Thermo Fisher AM2239), and 0.2 U/µL Superase RNAse Inhibitor (Thermo Fisher AM2696) in Ultrapure distilled water (Thermo Fisher 10977-015).

Gradient sucrose buffer: 10 or 45% w/v sucrose (Fisher 8510500GM), 20 mM Tris pH 7.5 (Thermo Fisher AM9850G, AM9855G), 100 mM NaCl (Thermo Fisher AM9760G), 15 mM $MgCl_2$ (Thermo Fisher AM9530G), 1 mM DTT (MilliporeSigma 43815), and 100 µg/mL cycloheximide (MilliporeSigma C7698) in Ultrapure distilled water (Thermo Fisher 10977-015).

A 10 cm plate of mESCs at ~50% confluence was treated with 100 µg cycloheximide (CHX) for 2 min at 37°C. Cells were washed, trypsinized, and neutralized with media as described above, except that all buffers contained 100 µg/mL CHX. The cells were pelleted at 200 × *g* × 3 min at 4°C, washed with ice-cold DPBS + CHX, and pelleted again. To the pellet, 400 µL gradient lysis buffer (above) was added and vortexed for 30 s with 30 s rest on ice for three cycles, then incubated for 30 min rotating at 4°C. The lysate was clarified by centrifuging at 700 × *g* for 5 min at 4°C, then again at 7000 × *g* for 5 min at 4°C. RNA concentration was measured by Nanodrop

2000 (Thermo Fisher), and samples were adjusted to equal concentrations using additional lysis buffer. 250 μL of clarified lysate was layered onto a 10–45% sucrose gradient (see above), which was made on a Biocomp Model 108 Gradient Master (Colorado Springs, CO). Gradients were spun on a Beckman SW-41 rotor (Indianapolis, IN) at 40,000 rpm for 2.5 hr at 4°C. After centrifugation, gradients were fractionated using a density gradient fraction system (Brandel BR-188, Gaithersburg, MD) with measurement of UV absorbance. Fractions were precipitated using the ProteoExtract protein precipitation kit (MilliporeSigma 539180) as per the manufacturer's protocol and redissolved in 2X Laemmli buffer (Fisher 50-196-784). For western blot, an equal volume of each fraction was loaded onto an SDS-PAGE gel.

## Cell culture western blot

Western blot lysis buffer: 25 mM Tris pH 7.5 (Thermo Fisher AM9850G, AM9855G), 150 mM NaCl (Thermo Fisher AM9760G), 15 mM $MgCl_2$ (Thermo Fisher AM9530G), 1% v/v Triton-X 100 (MilliporeSigma X100-500ML), 8% v/v glycerol (MilliporeSigma G6279), 1 mM DTT (MilliporeSigma 43815), 1X cOmplete mini protease inhibitor EDTA-free (Roche 11836170001), 0.5% w/v deoxycholate (MilliporeSigma S1827), 0.02 U/μL Turbo DNAse (Thermo Fisher AM2239), and 0.2 U/μL Superase RNAse Inhibitor (Thermo Fisher AM2696) in Ultrapure distilled water (Thermo Fisher 10977-015).

Unless otherwise stated, cells were washed, trypsinized, and neutralized with media as described above. The cells were pelleted at 200 × g for 3 min at 4°C, washed with ice-cold DPBS, pelleted again, and washed again in DPBS. Western blot lysis buffer (above) was added. Cells and lysis buffer were vortexed for 30 s and rested on ice for 30 s for three cycles, then incubated at 4°C for 15 min. Lysates were clarified by centrifuging at 7000 × g for 5 min at 4°C. Unless otherwise stated, total protein in each sample was quantified by bicinchoninic acid assay (Thermo Fisher 23225) as per the manufacturer's protocol and samples were normalized to equal total protein.

Samples were resolved on a 4–20% Tris-glycine gradient SDS-PAGE gel (Bio-Rad 5671095) and transferred to a polyvinylidene difluoride membrane (Bio-Rad 1704273). Membranes were blocked for 1 hr at room temperature with 5% milk in phosphate-buffered saline (PBS) (Fisher BP2944100) with 0.1% Tween-20 (MilliporeSigma P9416) (PBST). Blots were incubated for 16 hr at 4°C with the following primary antibodies at 1:1000 dilution in 5% BSA/PBST, unless stated otherwise: anti-Rps27 (1:10,000, Thermo Fisher PA5-18092), anti-Rps27l (ProteinTech 15871-1-AP, Rosemont, IL), anti-B-actin (Cell Signaling 3700S, Danvers, MA), anti-Rps3 (Abcam ab77330, Cambridge, UK), anti-FLAG (MilliporeSigma F3165), anti-Gapdh (1:5000, Thermo Fisher AM4300), and anti-Trp53 (Leica Biosystems CM5, Wetzlar, Germany). Membranes were washed three times for 10 min in PBST before incubation for 30 min at room temperature with secondary antibodies coupled to horseradish peroxidase at 1:10,000 dilution in 5% milk/PBST: donkey anti-mouse (GE Healthcare NA931-1ML, Chicago, IL), donkey anti-rabbit (GE Healthcare NA934-1ML), and chicken anti-goat (R&D Systems HAF019, Minneapolis, MN). Membranes were washed three times for 10 min in PBST before detection using Clarity Western ECL Substrate (Bio-Rad 170-5061) and imaging on a ChemiDoc MP (Bio-Rad 17001402). Protein band intensity was quantified using the Fiji distribution of ImageJ.

## Tissue western blot

Lysis buffer: 50 mM Tris-HCl pH 8 (Thermo Fisher AM9855G), 150 mM NaCl (Thermo Fisher AM9760G), 1% Triton X-100 (MilliporeSigma X100-500ML), 0.5% sodium deoxycholate (MilliporeSigma S1827), 0.1% sodium dodecyl sulfate (MilliporeSigma 436143), 1 mM EDTA (Thermo Fisher AM9260G), and 1X cOmplete Protease Inhibitor, EDTA-free (Roche 11836170001) in UltraPure distilled water (Thermo Fisher 10977-015).

After euthanasia as described above, mice were dissected. The relevant tissues were minced to 1 mm pieces, frozen in liquid nitrogen, and stored at –80°C. Chilled lysis buffer was added to the samples in 2 mL microcentrifuge tube. A handheld homogenizer (Fisher Scientific 150, 15-340-167) with a fresh 7 mm plastic probe was used on setting 4 for 10 s to homogenize each sample. Homogenized samples were incubated on ice for 30 min, then centrifuged at 17,000 × g for 10 min at 4°C. The supernatant was transferred to a fresh chilled tube. Normalization, denaturation, SDS-PAGE, western blotting, and quantification were performed as described above for cell culture western blots.

## Generation of CRISPR-edited mESCs

The CRISPR strategy for inserting 3xFLAG C-terminally at the endogenous *Rps27* or *Rps27l* locus in mESCs via exon replacement was designed as follows: first, two guide RNA (gRNA) recognition sites were identified that flanked the exon of *Rps27* or *Rps27l* containing the stop codon, using on- and off-target gRNA site scoring algorithms (*Doench et al., 2016*; *Hsu et al., 2013*) implemented in Benchling (San Francisco, CA). gRNA sites were only used if they had no other high-probability predicted cut sites throughout the mouse genome. Each 20 nt gRNA sequence was each cloned into a PX459 (*Ran et al., 2013*) (Addgene 62988, Watertown, MA) backbone digested with BbsI (Thermo FD1014), with a single upstream G nucleotide preceding the gRNA sequence since this has been shown to improve cutting efficiency (*Ran et al., 2013*). To construct the homology-directed repair template, the sequence between the two gRNA cut sites was cloned along with 300 bp homology arms on each end. Immediately preceding the stop codon in the repair template, a sequence was inserted to encode a 2xGGGS linker and a 3xFLAG peptide. The gRNA recognition sites or proto-spacer adjacent motifs (PAM) on the repair template were modified at silent coding positions or non-coding positions with low evolutionary conservation to prevent cutting of the repair template or of the repaired genomic DNA. To deliver the repair template, an unmodified PAM and gRNA recognition sequence were appended to each end of the repair template, distal to the homology arms. Distal to the appended gRNA and PAM sequences on each side, 10 additional bases were appended to buffer against small deletions that occur with TOPO cloning. This construct was then inserted into a non-expressing pCR4Blunt-TOPO backbone (Thermo Fisher 450031) such that the Cas9-gRNAs expressed from the PX459 plasmids would cleave the repair template as a linear dsDNA from the circular pCR4Blunt-TOPO plasmid (*Zhang et al., 2017a*).

At passage number 28, $10^6$ mESCs were transfected with the two PX459-based plasmids harboring a Cas9-puromycin fusion construct and gRNAs flanking the targeted exon (0.5 µg each), and one pCR4Blunt-TOPO-based plasmid harboring the linearizable repair template (2 µg) (see *Supplementary file 7* for sequences). The three combined plasmids were diluted in 100 µL Opti-MEM Reduced Serum Medium (Thermo Fisher 11058021). In parallel, 7.5 µL of Lipofectamine 2000 (Thermo Fisher 11668-019) was diluted in 100 µL Opti-MEM. The plasmid/Opti-MEM and Lipofectamine/Opti-MEM were combined and incubated for 20 min at room temperature. Cells were trypsinized as described above, resuspended in 250 µL Opti-MEM, added to the plasmid-Lipofectamine complexes for 10 min at room temperature, and plated into one 12-well. Media was changed after 4 hr. At 24 hr after transfection, cells were treated with media containing 1 µg/mL puromycin (Millipore P8833). At 48 hr after transfection, fresh media containing puromycin was changed in. At 72 hr after transfection, puromycin-free media was changed in. At 96 hr after transfection, cells were washed, trypsinized, dissociated, and plated at 1000 cells per 10 cm plate to form colonies derived from single cells. At 7 d after sparse plating, individual colonies were lifted using a pipet tip, dissociated at 37°C in 0.025% trypsin-EDTA in DPBS, replica plated in gelatinized 96-well plates in regular mESC media, and screened by genomic DNA PCR and western blot for desired edits. Importantly, PCR primers were designed such that at least one primer bound distal to the 300 bp homology arms to avoid amplifying residual repair template.

## FLAG-IP ribosome profiling

IP lysis buffer: 25 mM Tris pH 7.5 (Thermo Fisher AM9850G, AM9855G), 150 mM NaCl (Thermo Fisher AM9760G), 15 mM $MgCl_2$ (Thermo Fisher AM9530G), 1% v/v Triton-X 100 (MilliporeSigma X100-500ML), 8% v/v glycerol (MilliporeSigma G6279), 1 mM DTT (MilliporeSigma 43815), 1X cOmplete Mini Protease Inhibitor EDTA-free (Roche 11836170001), 0.5% w/v deoxycholate (MilliporeSigma S1827), 200 µg/mL cycloheximide (MilliporeSigma C7698), 0.02 U/µL Turbo DNAse (Thermo Fisher AM2239), and 0.2 U/µL Superase RNAse Inhibitor (Thermo Fisher AM2696) in Ultrapure distilled water (Thermo Fisher 10977-015).

Sucrose cushion buffer: 25 mM Tris pH 7.5 (Thermo Fisher AM9850G, AM9855G), 150 mM NaCl (Thermo Fisher AM9760G), 15 mM $MgCl_2$ (Thermo Fisher AM9530G), 1 mM DTT (MilliporeSigma 43815), 1X cOmplete Mini Protease Inhibitor EDTA-free (Roche 11836170001), and 200 µg/mL cyclo-heximide (MilliporeSigma C7698) in Ultrapure distilled water (Thermo Fisher 10977–015).

Wash buffer 1: Equivalent to IP lysis buffer but omitting glycerol, protease inhibitor, RNAse inhibitor, and DNAse.

Wash buffer 2: Equivalent to wash buffer 1, but with 300 mM NaCl.

Western elution buffer: 4% v/v SDS (MilliporeSigma 436143), 125 mM Tris pH 7.0 (Thermo Fisher AM9850G) in Ultrapure distilled water (Thermo Fisher 10977-015).

CRISPR-edited mESCs and control lines were grown to ~50% confluence in 2 × 15 cm plates. Three biological replicates (clones) were included for each genotype. Cells were treated with 100 µg/mL cycloheximide (CHX) for 2 min at 37°C, then washed twice with 20 mL DPBS containing 100 µg/mL CHX. Cells were scraped into 5 mL DPBS + CHX and centrifuged at 200 × $g$ for 3 min at 4°C, then washed again in DPBS + CHX. 800 µL IP lysis buffer without Superase RNAse Inhibitor was added to the pellet and vortexed for 30 s with 30 s rest for three cycles, then rotated at 4°C for 30 min. Lysate was clarified by centrifuging at 700 × $g$ for 5 min at 4°C, then again at 7000 × $g$ for 5 min at 4°C. RNA concentration was measured by Nanodrop 2000 (Thermo). By adding more IP lysis buffer without RNAse inhibitor, the clarified lysate was adjusted to a concentration of 1500 ng/µL in 600 µL total volume. 6 µL RNase A (Thermo EN0531) and 3.6 µL RNAse T1 (Thermo EN0541) were added, and the digestion reaction was rotated for 30 min at room temperature. RNAse digestion was quenched by placing it on ice and adding 30 µL of Superase RNAse Inhibitor. Two 250 µL aliquots of digested lysate were each layered over 750 µL of sucrose cushion buffer (see above). The cushions were ultracentrifuged in a TLA 120.2 rotor (Beckman, Indianapolis, IN) at 100,000 rpm for 1 hr at 4°C. Each pellet was rinsed once with 1 mL ice-cold Ultrapure water, then resuspended by pipetting and shaking in 250 µL IP lysis buffer containing Superase RNAse Inhibitor. An aliquot of the resuspended pellet was reserved in TRIzol (Thermo Fisher 15596-018) as the total RPF fraction. The two resuspended pellets from each replicate were combined and added to 200 µL mouse IgG beads (MilliporeSigma A0919), which had been equilibrated twice for 5 min each at 4°C in an equal volume of IP lysis buffer with RNAse inhibitor. The samples were pre-cleared with the IgG beads for 1 hr at 4°C on a rotator, then transferred to 200 µL anti-FLAG beads (MilliporeSigma A2220), which had been equilibrated twice for 5 min each at 4°C in an equal volume of IP lysis buffer with RNAse inhibitor. The immunoprecipitation reaction was rotated for 2 hr at 4°C, washed with 400 µL wash buffer 1 three times for 5 min each at 4°C, and washed with 400 µL wash buffer two three times for 5 min each at 4°C. For ribosome profiling, an aliquot of the beads was incubated in TRIzol for 5 min at room temperature and reserved as the IP RPF fraction. For western blot, an aliquot of the beads was heated with shaking to 95°C in western elution buffer (see above) for 5 min.

For western blot, an aliquot of the cushion supernatant and eluate were precipitated using the ProteoExtract protein precipitation kit (Calbiochem 539180) as per the manufacturer's protocol and redissolved in 2X Laemmli buffer (Fisher 50-196-784). 1% of the sample volume was loaded for lysate, supernatant, pellet, and flow-through samples. 10% of the sample volume was loaded for eluate samples.

Ribosome profiling libraries were prepared following the published protocol of *McGlincy and Ingolia, 2017* with modifications as stated below, using oligonucleotides synthesized by Integrated DNA Technologies (Coralville, IA; *Supplementary file 7*). Total RPFs and IP RPFs were extracted from TRIzol using the Direct-Zol Microprep Kit (Zymo R2060, Irvine, CA) according to the manufacturer protocol, omitting the DNAse digestion. RPF volume was adjusted to 90 µL with Ultrapure water. 10 µL 3 M NaOAc pH 5.5 (Thermo Fisher AM9740) and 2 µL of 15 mg/mL GlycoBlue (Thermo Fisher AM9515) were added before precipitating the RPFs in 150 µL 100% isopropanol overnight at –80°C. Precipitated RPFs were pelleted at 21,000 × $g$ for 30 min at 4°C, washed with ice-cold 80% ethanol in water, dried at room temperature for 10 min, and dissolved in Ultrapure water. RPFs were denatured at 80°C for 90 s in denaturing sample loading buffer (*McGlincy and Ingolia, 2017*), then incubated on ice for 5 min before running on a 15% Tris-borate-EDTA-urea (TBE-urea) polyacrylamide gel. Fragments were size-selected using NI-800 and NI-801 (*McGlincy and Ingolia, 2017*) as 26–34 nt markers. Gel slices were freeze-thawed for 30 min at –80°C, crushed, and extracted at room temperature overnight in 400 µL RNA extraction buffer (*McGlincy and Ingolia, 2017*), then re-extracted with an additional 200 µL RNA extraction buffer. The combined 600 µL extraction was precipitated with 2 µL GlycoBlue and 750 µL 100% isopropanol overnight at –80°C. The RPFs were pelleted, washed, and dried as described above, then dissolved in 4 µL of 10 mM Tris pH 8, dephosphorylated, and ligated to barcoded linkers as per the published protocol. Between replicates, barcodes were permuted among the samples. Unreacted linker was deadenylated and digested as per the published protocol. The barcoded RPFs were pooled within each replicate and purified on a Zymo Oligo Clean & Concentrator

column (Zymo D4060) according to the manufacturer's protocol. Pooled RPFs were diluted to 100 ng/µL and 1 µg was used as input for rRNA depletion using RiboZero Gold (part of Illumina 20020598, San Diego, CA) as per the manufacturer's protocol. rRNA-depleted RPFs were purified on a Zymo Oligo Clean & Concentrator column, then reverse-transcribed as per the published protocol. Template RNA was degraded by alkaline hydrolysis as per the published protocol. cDNA was purified on a Zymo Oligo Clean & Concentrator column, denatured in denaturing sample loading buffer, and size-selected on a 10% TBE-urea gel, as marked by NI-800 and NI-801 that had been ligated and reverse-transcribed in parallel with the samples. Gel slices were freeze-thawed for 30 min at –80°C, crushed, and extracted at room temperature overnight in 400 µL DNA extraction buffer (*McGlincy and Ingolia, 2017*), then re-extracted with an additional 200 µL DNA extraction buffer. The combined 600 µL extraction was precipitated with 2 µL GlycoBlue and 750 µL 100% isopropanol overnight at –80°C. The cDNA was pelleted, washed, and dried as described above, then resuspended in 15 µL 10 mM Tris pH 8. cDNA was circularized by adding 2 µL 10X CircLigase I buffer, 1 µL of 1 mM ATP, 1 µL of 50 mM MnCl$_2$, and 1 µL of CircLigase I (Lucigen CL4111K, Middleton, WI) to the 15 µL of cDNA and incubating at 60°C for 12 hr, then 80°C for 10 min. Circularized cDNA was purified on a RNA Clean & Concentrator column. cDNA concentration was quantified by qPCR as per the published protocol except using SsoAdvanced Universal SYBR Green qPCR master mix (Bio-Rad 1725274) and following the manufacturer's protocol. Library construction to add indexing primers was performed as per the published protocol using a different reverse primer for each replicate. PCR products were purified using a Zymo DNA Clean & Concentrator column (Zymo D4003). Size selection was performed on a 8% TBE-urea gel, with the lower bound marked by NI-803 (*McGlincy and Ingolia, 2017*) that had undergone library construction in parallel with the samples, and the upper bound at 170 nt as marked by O'Range 20 bp DNA ladder (Thermo SM1323). Gel slices were extracted as described above. DNA was precipitated as described above, except using 1.25 µL of 20 µg/mL glycogen (Themo 10814-010) instead of GlycoBlue in the precipitation and incubating at –80°C for 2 hr. The pellet was resuspended in 10 mM Tris pH 8. Library quality and concentration was analyzed on an Agilent 2100 Bioanalyzer (High-Sensitivity DNA) at the Stanford Protein and Nucleic Acid Facility. Libraries were sequenced by Novogene (Sacramento, CA) on an Illumina HiSeq 4000 with paired-end 150 bp reads.

## Ribosome profiling analysis

Analysis scripts are available on GitHub (see 'Data availability statement'). Due to the short insert length, only analysis of Read 1 was necessary. cutadapt version 2.4 (*Martin, 2011*) was used to trim 3' adapter sequences from Read 1 with parameters "-j 0 -u 3 -a AGATCGGAAGAGCACAGTCTGAAC TCCAGTCAC `--discard-untrimmed` -m 15". In-line barcodes were demultiplexed using fastx_barcode_splitter.pl (http://hannonlab.cshl.edu/fastx_toolkit/) with parameters "--eol". Unique molecular identifiers and in-line barcodes were extracted using umi_tools version 1.0.1 (*Smith et al., 2017*) with parameters "extract `--extract-method=string --bc-pattern=NNNNNCCCCC --3prime`". Reads were filtered by quality using fastq_quality_filter (http://hannonlab.cshl.edu/fastx_toolkit/) with parameters "-Q33 -q 20 -p 70 -z". To remove reads originating from rRNA, transfer RNA (tRNA), and small nuclear RNA (snRNA), reads aligning to these sequences using bowtie2 version 2.3.4.3 (*Langmead and Salzberg, 2012*) with parameters "-L 18" were discarded. Remaining reads were aligned using bowtie2 with parameters "--norc -L 18" to a reference GRCm38/mm10 mouse transcriptome that was derived from UCSC/GENCODE VM20 knownCanonical annotations filtered for transcripts associated with at least one of the following: a Uniprot ID, a RefSeq ID, or an Entrez ID. PCR duplicates were removed using umi_tools. RPFs were parsed for uniquely aligned reads and grouped by read length. Ribosome A site positions were determined by offsetting the distance of the 5' end of each read to canonical start sites in each length group and adding 4 nucleotides. RPF reads aligning to the coding sequence (CDS) of a transcript (excluding the first 15 codons and last 5 codons of each CDS) were counted using the above transcriptome annotation. Because genes encoded by mitochondrial DNA are translated by mitoribosomes in the mitochondrial lumen that are distinct from cytoplasmic ribosomes, reads mapping to these genes were excluded from further analysis. Transcripts with counts per million (CPM) >2 were retained for downstream analysis. IP RPF and total RPF libraries were normalized separately by the trimmed mean of M-values method in edgeR (*Robinson et al., 2010*).

Differential RPF abundance and enrichment were analyzed using voom (*Law et al., 2014*) and limma (*Ritchie et al., 2015*). The following terminology describes the variables included in the analysis:

'cell line' distinguishes *Rps27-FLAG*, *Rps27l-FLAG*, and WT mESCs; 'biological replicate' distinguishes each mESC clone (three independently selected clones per cell line); and 'fraction' distinguishes IP and total RPFs.

To identify transcripts that were differentially abundant between the total RPFs in *Rps27-FLAG*, *Rps27l-FLAG*, and WT mESCs, the following design matrix was used: '~0 + cell line.' Contrast matrices were constructed for each pairwise comparison: *Rps27-FLAG* versus WT, *Rps27l-FLAG* versus WT, and *Rps27l-FLAG* versus *Rps27-FLAG*.

To identify transcripts that were differentially abundant between the IP and total RPFs of each cell line, the following design matrix was used: '~0 + fraction.' Contrast matrices were constructed for each pairwise comparison: Rps27-FLAG IP versus total, Rps27l-FLAG IP versus total.

To identify transcripts that were differentially enriched by anti-FLAG IP from among the total RPFs in *Rps27-FLAG* versus *Rps27l-FLAG* mESCs, the following design matrix was constructed: '~0 + cell line + cell line:biological replicate + cell line:fraction.' The following comparison was made: Rps27l-FLAG:IP versus Rps27-FLAG:IP.

For all comparisons, the data for the relevant samples were transformed using voom to remove mean-variance count heteroscedasticity. Using the respective design and contrast matrices described above, linear models were fitted to the data using limma with empirical Bayes moderation. Transcripts with significant differential RPF abundance or enrichment were defined as those with false discovery rate <0.05 obtained using the Benjamini–Hochberg method.

Gene Ontology enrichment analysis was performed using the goana function in limma after excluding a *Tmod3* transcript as a likely artifact of anti-FLAG IP (Appendix 1, *Appendix 1—figure 1*).

## Generation of genetically edited mouse lines

The CRISPR strategy to produce mice with *Rps27* or *Rps27l* homogenization or truncation is analogous to the strategy described above for CRISPR insertion of 3xFLAG into mESCs. Highly specific gRNA recognition sites were selected that flanked exon 2 of either gene, which encodes the three residues that differ between the paralogs. Instead of a linearizable dsDNA template, an ssDNA repair template with 100 nt homology arms was synthesized by GeneWiz (South Plainfield, NJ). This repair template contained the modified codons to homogenize the targeted paralog. Additionally, the gRNA recognition sites or protospacer adjacent motifs (PAM) on the ssDNA repair template were modified at silent coding positions or non-coding positions with low evolutionary conservation to prevent cutting of the repaired genomic DNA. gRNA and ssDNA sequences are listed in *Supplementary file 7*.

CRISPR-edited mouse lines were generated at the Gladstone Institute Transgenic Gene Targeting Core (San Francisco, CA). All animal procedures were approved by the Institutional Animal Care and Use Committee at the University of California, San Francisco (protocol #AN180952-01B). Superovulated female C57BL/6 mice (4 weeks old) were mated to C57BL/6 stud males. Fertilized zygotes were collected from oviducts and injected with Cas9 protein (20 ng/μL), two sgRNAs with recognition sites that flanked exon 2 (10 ng/μL each) (synthesized by IDT as Alt-R CRISPR-Cas9 crRNA, Coralville, IA), and one ssDNA repair template (10 ng/μL) into the pronucleus of fertilized zygotes. Injected zygotes were implanted into oviducts of pseudopregnant CD1 female mice. The targeted loci were PCR-amplified from genomic DNA of F0 animals, subcloned, and Sanger sequenced to identify successfully edited alleles. The *Rps27^{exon2del}* and *Rps27l^{exon2del}* alleles were recovered serendipitously from mice injected with reagents designed to produce the *Rps27^{Rps27l}* and *Rps27l^{Rps27}* alleles, which were also successfully recovered. F0 mice were backcrossed to wild-type C57BL6 males and females (Jackson Labs) for at least four generations.

## Adult mouse necropsy and bloodwork

For necropsy of 10–16-week-old adult male mice, mice were euthanized by $CO_2$ asphyxiation and cardiac exsanguination. Whole cardiac blood was collected in EDTA-coated microtainers (Becton Dickinson 365974) and 1.5 mL plastic tubes for complete blood counts and serum biochemistry, respectively. Blood samples were analyzed as described below. Mice were routinely processed for gross examination. In addition to body weight, the following organs were weighed, and a percentage of body weight calculation was conducted for each organ: liver, spleen, heart, kidneys (left and right), and testicles (left and right). Tissues were immersion-fixed in 10% neutral buffered formalin (Fisher Scientific) for 72 hr. Tissues containing bone were decalcified for 24 hr using Cal-Ex II Fixative/

Decalcifier (Fisher CS511-1D). Formalin-fixed tissues were processed routinely, embedded in paraffin, sectioned at 5 µm, and stained with hematoxylin and eosin. The following organs were evaluated histologically by a board-certified veterinary pathologist blinded to the sample genotypes: liver, kidneys, heart, spleen, thymus, pancreas, salivary glands, lungs, thyroid, trachea, esophagus, tongue, haired skin (interscapular), testes, accessory sex glands (preputial, seminal vesicles, prostate), urinary bladder, brain, gastrointestinal tract, bone marrow (pelvic limb), nasal cavity, eyes, teeth, ears, vertebral column, and spinal cord.

For bloodwork of aged adult mice, male mice were co-housed with male littermate controls until 9–10 months of age. Mice were euthanized by $CO_2$ asphyxiation and cardiac exsanguination. Whole cardiac blood was collected as described above. Samples were handled at room temperature during collection and submitted for analysis within 1 hr of collection. Automated hematology was performed on a Sysmex XN-1000V hematology analyzer. Blood smears were made for all CBC samples, Wright–Giemsa stained, and reviewed by a clinical laboratory scientist. Manual differentials were performed as indicated by species and automated analysis. Liver panel analysis was performed on a Siemens Dimension EXL200/LOCI analyzer.

## Mammary carmine alum, H&E

After euthanasia, mammary glands were dissected from littermate-matched female mice at the indicated pregnancy and lactation stages. From each mouse, the left abdominal mammary gland was used for carmine alum staining as previously described (*Plante et al., 2011*). The right abdominal mammary gland was fixed overnight in 10% neutral-buffered formalin, washed in phosphate buffered saline with 0.2% w/v glycine, and stored in 70% ethanol until paraffin embedding, sectioning, and staining with hematoxylin-eosin.

## Mouse embryonic fibroblast isolation and culture

Mouse embryonic fibroblasts (MEFs) were isolated at E13.5 as previously described (*Durkin et al., 2013*) from heterozygous crosses of the $Rps27^{Rps27l}$ and $Rps27l^{Rps27}$ lines. MEFs were cultured in DMEM with high glucose (Gibco 11965), supplemented with 10% fetal bovine serum (MilliporeSigma TMS-013-B) and 1% penicillin/streptomycin (Thermo Fisher 15140-122). MEFs were cultured at 37°C with 5% $CO_2$ and passaged every 2–3 d following the same passaging protocol as performed for mESCs (above). MEFs were used for experiments at passage numbers 2–4.

## EdU/DAPI flow cytometry

MEFs were plated at 250,000 cells per T25 flask. After 72 hr, 5-ethynyl-2'-deoxyuridine (EdU) (Click Chemistry Tools 1381, Scottsdale, AZ) was added to a final concentration of 10 µM. The media was kept warm during the EdU addition and thoroughly mixed in the flask afterward. The cells were incubated at 37°C for 2 hr, then washed once with 3 mL of 0.05% trypsin-EDTA in DPBS and trypsinized for 5 min at 37°C in 1 mL 0.05% trypsin-EDTA in DPBS. The trypsin was neutralized with 1 mL MEF media and pipetted thoroughly to obtain a single-cell suspension. The media and trypsin wash previously poured off from the flask was recombined with the cells. The cells were pelleted at 200 × *g* for 3 min at room temperature and washed in 1 mL 1% bovine serum albumin (BSA) in DPBS. For each MEF line, 500K cells were fixed in 200 µL 4% paraformaldehyde in PBS (dilution of Fisher 43368-9M in 1.33× PBS) for 15 min in the dark at room temperature. To the fixative reaction, 1 mL ice-cold 70% ethanol was added and incubated on ice for 10 min. Cells were washed twice in 1 mL DPBS + 0.1% Triton X-100 and resuspended in 50 µL DPBS + 0.1% Triton X-100. The click reaction master mix was prepared as per the manufacturer's instructions (Click Chemistry Tools 1381) and 250 µL was added to each sample. The click reaction was incubated for 30 min in the dark at room temperature, then washed with 1 mL DPBS + 0.1% Triton X-100. The cells were resuspended in 150 µL DPBS + 0.1% Triton X-100 containing 4 µg/mL 4',6-diamidino-2-phenylindole (DAPI) (Thermo 62248), filtered through a mesh-top tube, and incubated in the dark overnight at 4°C before running on a Novocyte Quanteon flow cytometer using NovoExpress 1.3.0 software (Agilent, Santa Clara, CA) at a flow rate of 14 µL/min. Gating for singlets was performed in FlowJo (Ashland, OR) on forward scatter area (FSC-A) × forward scatter height (FSC-H), then DAPI area × height.

## Cell viability assays

MEFs were plated in the appropriate media (see above) in 96-well half-area black plates (Corning CLS3603) at 2500 MEFs per well for 24 hr prior to treatment. Media was exchanged for media containing doxorubicin (MilliporeSigma 324380) or etoposide (MilliporeSigma E1383) at the specified doses. Cells were incubated for the specified time. Cell viability was measured using CellTiterGlo 2.0 (Promega G9242) following the manufacturer's instructions.

## Statistical analysis

Sample sizes, the number of technical or biological replicates, statistical tests, significance values, and significance thresholds are reported in the main text or figure legends pertaining to each experiment. No explicit power analysis was used to predetermine sample size. Randomization was not applicable for these experiments. No samples were excluded from analysis.

## Acknowledgements

We thank the members of the Barna, Pritchard, and Attardi labs for advice and critiques; Yiran Liu for advice on designing CRISPR strategies; Elias Godoy for assistance with necropsy dissections; Jared Dunnmon for advice on computational resources; the Stanford Animal Histology Core for assistance with histology sample preparation; and the Stanford Animal Diagnostic lab for preparation of histology samples. We thank the Stanford Shared FACS Facility for assistance with flow cytometry experiments. We thank the UCSF Gladstone Institutes Transgenic Gene-Targeting Core for facility support in generating CRISPR-edited mice. pSpCas9(BB)-2A-Puro (PX459) V2.0 was a gift from Feng Zhang (Addgene plasmid # 62988; RRID:Addgene_62988).

## Additional information

### Funding

| Funder | Grant reference number | Author |
|---|---|---|
| National Institutes of Health | F30HD100123 | Adele Francis Xu |
| Stanford Bio-X | | Adele Francis Xu |
| National Institutes of Health | 5R01HG008140 | Jonathan K Pritchard |
| New York Stem Cell Foundation | NYSCF-R-I36 | Maria Barna |
| National Institutes of Health | R01HD086634 | Maria Barna |
| Alfred P. Sloan Foundation | | Maria Barna |
| Pew Charitable Trusts | | Maria Barna |
| National Institutes of Health | R01HD098722 | Lindsay Hinck |

The funders had no role in study design, data collection and interpretation, or the decision to submit the work for publication.

### Author contributions

Adele Francis Xu, Conceptualization, Formal analysis, Investigation, Writing – original draft; Rut Molinuevo, Resources, Formal analysis, Supervision, Investigation, Writing – original draft, Writing – review and editing; Elisa Fazzari, Resources, Writing – review and editing; Harrison Tom, Conceptualization, Resources, Supervision, Writing – review and editing; Zijian Zhang, Conceptualization, Formal analysis, Supervision, Funding acquisition, Writing – review and editing; Julien Menendez, Kerriann M Casey, Formal analysis, Investigation, Writing – original draft; Davide Ruggero, Resources, Supervision, Writing – review and editing; Lindsay Hinck, Conceptualization, Supervision, Writing – review

and editing; Jonathan K Pritchard, Conceptualization, Formal analysis, Supervision, Writing – review and editing; Maria Barna, Conceptualization, Supervision, Funding acquisition, Investigation, Writing – review and editing

### Author ORCIDs
Adele Francis Xu http://orcid.org/0000-0003-3332-5285
Kerriann M Casey http://orcid.org/0000-0003-4228-928X
Davide Ruggero http://orcid.org/0000-0002-9444-5865
Jonathan K Pritchard http://orcid.org/0000-0002-8828-5236
Maria Barna http://orcid.org/0000-0002-6843-4396

### Ethics
All animal work was reviewed and approved by the Stanford Administrative Panel on Laboratory Animal Care (APLAC, protocol #27463). The Stanford APLAC is accredited by the American Association for the Accreditation of Laboratory Animal Care. All mice used in the study were housed at Stanford University except where otherwise noted. CRISPR-edited mouse lines were generated at the Gladstone Institute Transgenic Gene Targeting Core (San Francisco, CA). All animal procedures were approved by the Institutional Animal Care and Use Committee at the University of California, San Francisco (protocol #AN180952-01B).

### Decision letter and Author response
Decision letter https://doi.org/10.7554/eLife.78695.sa1
Author response https://doi.org/10.7554/eLife.78695.sa2

## Additional files

### Supplementary files
• MDAR checklist

• Supplementary file 1. Details of the Rps27 and Rps27l ortholog protein sequences used for multispecies alignment and molecular phylogenetic analysis in *Figure 1* and *Figure 1—figure supplement 1*.

• Supplementary file 2. Correlation between Rps27l mRNA abundance (read count per 100 RP reads) and abundance of other transcripts (read count per million) in single-cell RNA-seq data *Bach et al., 2017* from alveolar cells in lactating mammary glands ('Avd-L' cells as termed by Bach et al.).

• Supplementary file 3. Results of paralog-specific Rps27- and Rps27l-FLAG ribosome profiling.

• Supplementary file 4. Organ weights and necropsy findings for 10–16-week-old mice homozygous for homogenized Rps27$^{Rps27l}$ and Rps27l$^{Rps27}$ alleles. WNL, within normal limits.

• Supplementary file 5. Biomarker values for complete blood count, liver panel, and serum chemistry panel performed on 10–16-week-old homozygous males and age-matched wild-type controls from the Rps27$^{Rps27l}$ and Rps27l$^{Rps27}$ mouse lines.

• Supplementary file 6. Biomarker values for complete blood count and liver panel performed on homozygous and heterozygous 9–10-month-old males and age-matched wild-type controls from the Rps27$^{Rps27l}$ and Rps27l$^{Rps27}$ mouse lines.

• Supplementary file 7. Primer, oligonucleotide, and construct sequences used in this study.

### Data availability
Ribosome profiling sequencing data have been deposited in GEO under accession code GSE201845. Other data generated in this study are provided in the supplementary materials and source data files. Code used for data analysis is available at https://github.com/adelefxu/eS27_paralogs (copy archived at *Xu, 2023*).

The following dataset was generated:

| Author(s) | Year | Dataset title | Dataset URL | Database and Identifier |
|---|---|---|---|---|
| Barna M, Xu AF | 2023 | Subfunctionalized expression drives evolutionary retention of ribosomal protein paralogs Rps27 and Rps27l in vertebrates | https://www.ncbi.nlm.nih.gov/geo/query/acc.cgi?acc=GSE201845 | NCBI Gene Expression Omnibus, GSE201845 |

The following previously published datasets were used:

| Author(s) | Year | Dataset title | Dataset URL | Database and Identifier |
|---|---|---|---|---|
| Han X, Guo G | 2018 | MCA DGE Data | https://doi.org/10.6084/m9.figshare.5435866.v8 | figshare, 10.6084/m9.figshare.5435866.v8 |
| Webber J, Batson J, Pisco A, Tabula Muris Consortium | 2018 | Single-cell RNA-seq data from Smart-seq2 sequencing of FACS sorted cells (v2) | https://doi.org/10.6084/m9.figshare.5829687.v8 | figshare, 10.6084/m9.figshare.5829687.v8 |
| Bach K, Pensa S, Grzelak M, Hadfield J | 2017 | Differentiation dynamics of mammary epithelial cells revealed by single-cell RNA-sequencing | https://www.ncbi.nlm.nih.gov/geo/query/acc.cgi?acc=GSE106273 | NCBI Gene Expression Omnibus, GSE106273 |
| Rios AC, Pal B, Soetanto R, Fu NY | 2015 | Transcriptome analysis of luminal and basal cell subpopulations in the lactating versus pregnant mammary gland | https://www.ncbi.nlm.nih.gov/geo/query/acc.cgi?acc=GSE60450 | NCBI Gene Expression Omnibus, GSE60450 |

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

## Appendix 1

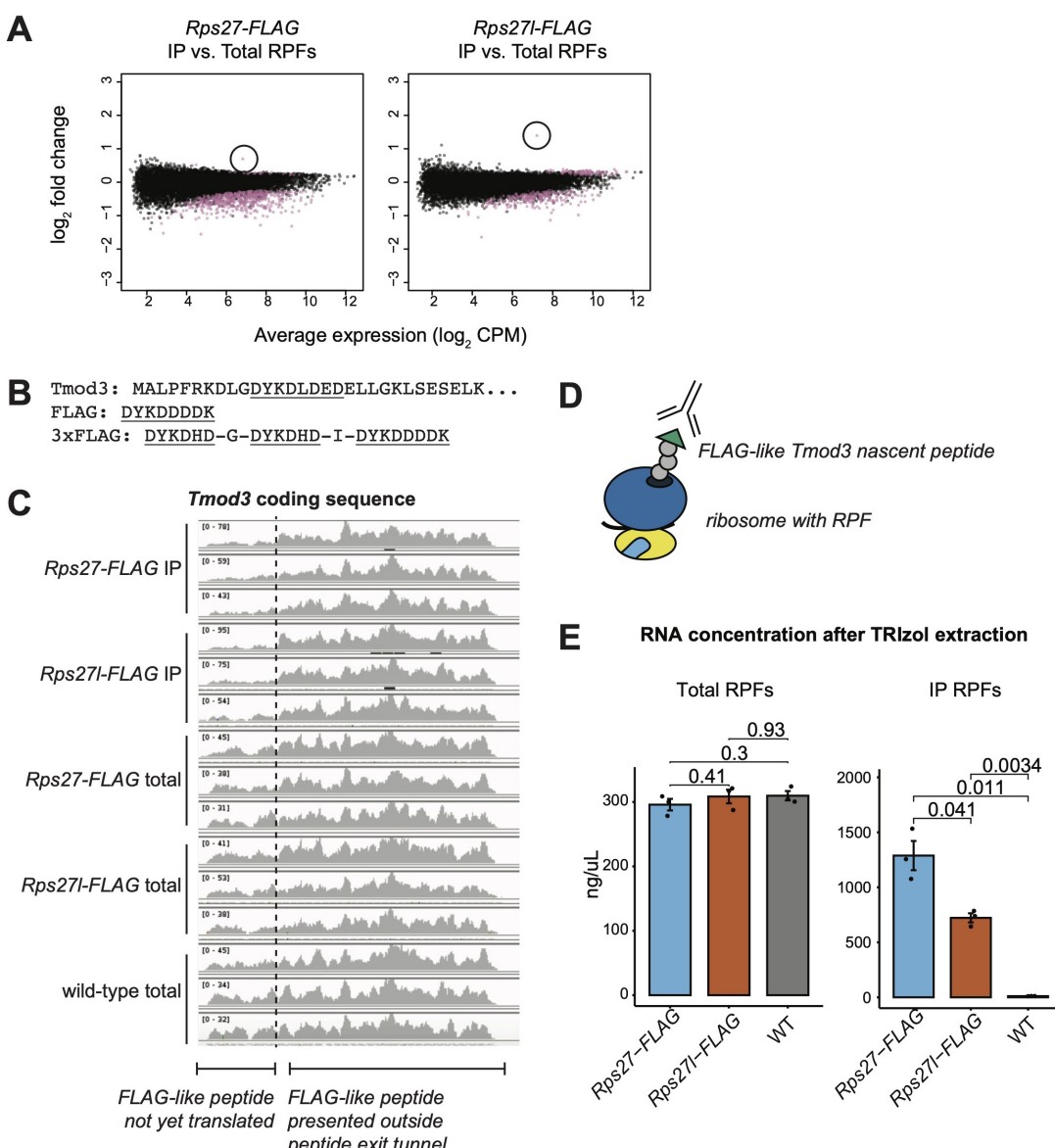

**Appendix 1—figure 1.** Enrichment of *Tmod3* ribosome-protected fragments is an artifact of FLAG immunoprecipitation. (**A**) Enrichment of *Tmod3* mRNA (circled) among immunoprecipitated ribosome-protected fragments (IP RPFs) from both *Rps27*- and *Rps27l-FLAG* mouse embryonic stem cells (mESCs), relative to total RPFs. (**B**) N-terminal sequence of *Tmod3* protein, with FLAG-like peptide sequence highlighted. (**C**) Distribution of IP RPFs and total RPFs along the coding sequence of *Tmod3*. Base positions encoding the FLAG-like peptide are denoted with an arrow. Note that the y-axis range differs for each sample. (**D**) Proposed binding of the anti-FLAG antibody to the FLAG-like *Tmod3* nascent peptide. (**E**) RNA concentration as estimated by absorbance at 260 and 280 nm wavelength after TRIzol extraction of total RPF and IP RPF samples from *Rps27-FLAG*, *Rps27l-FLAG*, and WT mESCs, showing consistently lower IP yield of Rps27l-ribosomes compared to Rps27l-ribosomes. Significance was assessed by *t*-test. n = 3 biological replicates. Error bars show standard error.

## FLAG-IP ribosome profiling artifact

Here we present evidence that a *Tmod3* transcript detected during ribosome profiling of Rps27- and Rps27l-ribosomes (*Figure 3D–F*) is an artifact of the FLAG-IP ribosome profiling process, and not likely a bona fide Rps27l-ribosome-associated transcript. Comparing IP RPFs to total RPFs for

both the *Rps27-FLAG* and *Rps27l-FLAG* mESC lines, we detected that *Tmod3* mRNA was more enriched than other transcripts in the IP RPFs for both lines. This enrichment was more pronounced for the *Rps27l-FLAG* line than for *Rps27-FLAG* (*Appendix 1—figure 1A*). Interestingly, Tmod3 protein is frequently reported as a contaminant of FLAG IPs in the mass spectrometry CRAPome, a repository of proteins that are non-specifically detected across many affinity purification experiments with different baits (*Mellacheruvu et al., 2013*). We inspected the amino acid sequence of *Tmod3* and found that it contains a sequence with high homology to the FLAG and 3xFLAG epitope tag sequences (*Appendix 1—figure 1B*). Furthermore, we found that Rps27- and Rps27l-FLAG IP RPFs are relatively enriched in *Tmod3* reads at the 3′ end of the *Tmod3* coding sequence relative to the 5′ end. The beginning of this enriched interval corresponds with the base position in the coding sequence at which the ribosome would have completely translated *Tmod3's* N-terminal FLAG-like sequence and presented it as a nascent peptide chain emerging from the ribosome's peptide exit tunnel (*Appendix 1—figure 1C*). We thus propose that ribosomes that are translating *Tmod3* transcripts are preferentially enriched during FLAG IP, whether or not they contain Rps27-FLAG or Rps27l-FLAG (*Appendix 1—figure 1D*). We also note that the RNA concentration of IP RPFs obtained from *Rps27-FLAG* mESCs was greater than that of *Rps27l-FLAG* mESCs (*Appendix 1—figure 1E*). This suggests that a larger proportion of anti-FLAG binding sites remained unoccupied during the Rps27l-FLAG IP compared to the Rps27-FLAG IP, and may explain why *Tmod3* appears to be more enriched among Rps27l-FLAG IP RPFs.

