## [Editor Report]

This article focuses on the fate of two ribosomal protein genes, *Rps27* and *Rps27L*, of vertebrates after they split in whole-genome duplication. The major strength is the support from solid laboratory experiments and an evolutionary perspective. It is a valuable case study revealing the differentiated roles in protein synthesis and expression patterns of duplicated genes.

---

## [Decision Letter]

**Decision letter after peer review:**

Thank you for submitting your article "Subfunctionalized expression drives evolutionary retention of ribosomal protein paralogs in vertebrates" for consideration by *eLife*. Your article has been reviewed by 3 peer reviewers, one of whom is a member of our Board of Reviewing Editors, and the evaluation has been overseen byGeorge Perry as the Senior Editor. The following individual involved in review of your submission has agreed to reveal their identity: Jonathan Dinman (Reviewer #3).

Essential revisions:

1) As pointed out in item 1 by Reviewer #2, the authors should consider performing western blotting experiments to confirm the amount of protein products should make the manuscript more convincing.

2) As pointed out by Reviewer #1 and Reviewer #3 (item 1), the authors should consider improving phylogenetic analysis and comparative data presentation to provide a more reliable timing and mode of the gene duplication eS27 and eS27L.

*Reviewer #1 (Recommendations for the authors):*

I would basically congratulate the authors for achieving so thorough experiments and data analysis, but have two crucial comments.

Most critically, the authors need to present molecular phylogenetic tree and synteny conservation data to show the timing of the gene duplication between eS27 and eS27L, rather than just speculate on it. Also, their alignment in Figure 1A and the description about evolutionary conservation of any kind should ideally include non-tetrapod vertebrates, such as zebrafish, spotted gar, sharks, and jawless fishes, even if they cannot produce any data from laboratory experiments using these animals. It is very unbalanced that the authors invested so much on functional characterizations of the two proteins, but pay little attention to molecular phylogeny which should be the basis of the study.

The title of the manuscript should include the specific names of the target genes of the study.

*Reviewer #3 (Recommendations for the authors):*

1. Figure 1A: Conclusion from Figure 1A is that the eS27 gene duplication most likely occurred through a DNA-based gene duplication event in an ancestor common to most vertebrates. However, in this figure, the alignments are mostly mammals, with only one bird and one amphibian represented. Given that, in the end, this is a paper about evolutionary biology, they should go deeper into the phylogenetic analysis. In particular, they should more deeply examine the junction between vertebrates and invertebrates. For example, in amphibians, is S27 also duplicated in salamanders? In Caecillians? And in fish, they should perform a wider analysis, from the most "recent", e.g. lungfish and coelacanths, down to sharks, hagfish and lamprey. It would also be interesting to examine a few advanced invertebrates such as sea lilies, starfish and sea urchins.

2. Figure 3B, closeup insert: what is the identity of the SSU protein that the eS27 amino acid residues 5, 12, and 17 about? Might this be important? At the very least, this RP should be labeled.

3. I found it interesting that both liver and lactating mammary glands preferentially contain eS27L enriched ribosomes as compared to heart, brain and nulliparous mammary gland. I have two observations regarding this. First, both are organs that tend to undergo higher amounts of cell replication compared to heart, brain and NP mammary gland. This is consistent with the observation of higher association of eS27L ribosomes with cell cycle-associated mRNAs. Testing whether ribosomes are reprogrammed to contain eS27L in highly dividing versus stationary phase cultured cells would be a relatively quick and inexpensive way to test this idea. Secondly, these two organs also produce large amounts of secreted proteins as compared to the other organs. Given that the group has tagged versions of the two proteins, they could test whether or not eS27L ribosomes are preferentially associated with endoplasmic reticulum than their eS27 counterparts using cultured cells. Alternatively, they might examine whether ribosomes are reprogrammed to contain eS27L when cells are programmed to become secretory cells, e.g. compare resting B-cells to B-cells stimulated to secrete antibody (i.e. plasma cells). I note that these experiments are mere suggestions, and are not required.

4. In a similar vein, rapidly duplicating and secreting cells such as liver and lactating mammary glands synthesize more protein than brain, heart and NP mammary gland cells. Perhaps the eS27L form produces ribosomes that are better tuned for speed versus accuracy. This can be tested by measuring rates of protein synthesis differences in translational accuracy in eS27 and eS27L homogenized cell lines.

5. Although it is clear that there are no differences between the homogenized mice and their heterologous littermates, one possible experiment would be to examine whether homogenization is disfavored over the course of multiple generations, i.e. a long term fitness experiment. However, given the extremely high costs and potentially low impact of such an experiment, I do not recommend that it be done. This question might better be performed in another, less expensive model organism such as *C. elegans* or *D. melanogaster* using a different paralogous pair of RP genes.

6. p. 13, line 14: Sentence beginning with 'nevertheless…' is superfluous.

---

## [Author Response]

Essential revisions:1) As pointed out in item 1 by Reviewer #2, the authors should consider performing western blotting experiments to confirm the amount of protein products should make the manuscript more convincing.

We agree that this additional data will strengthen the characterization of our mouse models. We have now performed Western blots for Rps27 and Rps27l protein and Gapdh loading control on liver, heart, forelimb, and brain tissue of postnatal day 0 mice that are wild-type, heterozygous, or homozygous for the Rps27^exon2del^ and Rps27l^exon2del^ truncation alleles. We did not include the Rps27^exon2del / exon2del^ genotype from these blots because we have previously shown that it is early embryonic lethal (revised Figure 4C-D) and the majority of these embryos yield no recoverable material even when dissecting as early as embryonic day 8.5. These Western blots and accompanying quantification of protein amounts are shown in Figure 4E-F, and are discussed in the highlighted text additions to the section “Rps27 and Rps27l homozygous knockouts are lethal at different developmental stages.” In brief, the results confirm that detectable protein expression from the respective truncated paralog is slightly diminished in Rps27^exon2del / +^ specimens and nearly eliminated in Rps27l^exon2del / exon2del^ specimens, with residual detection due to trace antibody cross-reactivity against the intact paralog (as demonstrated in Figure 3—figure supplement 1A, D). Interestingly, we observed that Rps27l^exon2del / +^ and Rps27l^exon2del / exon2del^ tissues had increased expression of the intact Rps27 protein, which is analogous to previously reported instances in which depletion of an RP gene is associated with increased expression of a paralogous gene (O’Leary et al. 2013; Milenkovic et al. 2023). Despite the increase in Rps27 protein level, Rps27l^exon2del / exon2del^ is early postnatal lethal; this suggests that the quantity of Rps27 increase in mice that are homozygous for the Rps27l^exon2del^ allele is insufficient to compensate for the Rps27l deficiency.

We have also performed Western blots for Rps27 and Rps27l protein and Gapdh loading control on tissue samples from adult mice that are wild-type, heterozygous, or homozygous for the Rps27^Rps27l^ and Rps27l^Rps27^ homogenized alleles. These Western blots and accompanying quantification of protein amounts are shown in Figure 5D-E, and are discussed in the highlighted text additions to the section “Rps27 and Rps27l proteins are functionally interchangeable across all examined murine tissues.” In brief, these new results confirmed loss of Rps27 protein and increased Rps27l protein in Rps27^Rps27l / Rps27l^ mice. This effect is most pronounced in the spleen, which contains abundant B cells, one of the cell types that highly express the Rps27 locus as shown above (Figure 1D). Likewise, loss of Rps27l protein expression with concurrently increased Rps27 protein was observed in Rps27l^Rps27 / Rps27^ mice and the effect is most pronounced in liver tissue, which contains hepatocytes that highly express the Rps27l locus.

We have also added the methods and source data for these Western blots to the methods section and source data zipped file, respectively.

2) As pointed out by Reviewer #1 and Reviewer #3 (item 1), the authors should consider improving phylogenetic analysis and comparative data presentation to provide a more reliable timing and mode of the gene duplication eS27 and eS27L.

We agree that additional analysis of Rps27’s evolutionary history would improve the robustness of our conclusions regarding its origin. First, following the recommendation of Reviewer #1 to present synteny conservation data, we now depict a 5 Mb window surrounding the Rps27 and Rps27l loci in the human genome and annotate several nearby genes that form a syntenic block between the two loci (Figure 1A). We previously described the similar exon structure between Rps27 and Rps27l, and we now depict this as well in Figure 1—figure supplement 1A. Together, the synteny and shared exon structure strongly support that the duplication occurred through a DNA-based mechanism such as DNA transposition or tandem/segmental/whole-genome duplication, as opposed to retrotransposition of RNA which generally results in intronless processed pseudogenes and would not include surrounding genes.

Regarding the timing of the duplication, we have followed the recommendations of Reviewers #1 and #3 to expand the multispecies protein sequence alignment, now presented in Figure 1C with the addition of newt, caecilian, lungfish, coelacanth, zebrafish, gar, shark, lamprey, sea squirt, amphioxus, sea star, sea urchin, and worm to represent more amphibians, fish, cyclostomes, and invertebrates. We also present a phylogenetic tree showing these and additional species (sponge, limpet, crinoid (closely related to sea lilies), hagfish, lizard, carp, pufferfish, tilapia, and salmon) with the putative timings of the 1R – 4R whole-genome duplications marked, and the number of Rps27 paralogs per species displayed (Figure 1B). We also constructed a molecular phylogenetic tree based on coding sequences of the Rps27/Rps27l orthologs across multiple species (Figure 1—figure supplement 1B). The results are discussed in the section “Rps27 and Rps27l are vertebrate ohnologs encoding highly conserved proteins.” In brief, while none of the nine included invertebrates had more than one Rps27 ortholog, nearly all vertebrates had two Rps27 paralogs, and most teleost fish species had three to six. These increases in Rps27 copy number thus coincide with the proposed timing of several WGDs. Furthermore, the Rps27 sequences from mammals and coelacanth formed a distinct clade from the corresponding Rps27l sequences, suggesting that duplication and subsequent divergence of these loci began in a common ancestor of these species.

We are aware that the 2R hypothesis has been challenged in recent years (i.e. Smith et al. 2018) and the WGD history of cyclostomes (hagfish and lampreys) remains controversial (i.e. Simakov et al. 2020, Nakatani et al. 2021). A detailed discussion of Rps27 paralogs in cyclostomes is beyond the scope of this work, but we did detect two Rps27 paralogs in lamprey and one in hagfish; the latter may be due to incomplete genome assembly/annotation or lineage-specific loss of a paralog. We also found that the confidence of the CDS phylogenetic tree was limited by the relatively short CDS and high conservation of Rps27 and the substantial evolutionary distance between the included species; nevertheless, these lines of evidence overall strongly suggest that the present-day Rps27 paralog pair is a remnant of the 1R/2R WGDs in the common vertebrate ancestor.

Details on the reference genomes and sequences used for phylogenetic analysis are available in Supplementary File 1.

Reviewer #1 (Recommendations for the authors):I would basically congratulate the authors for achieving so thorough experiments and data analysis, but have two crucial comments.Most critically, the authors need to present molecular phylogenetic tree and synteny conservation data to show the timing of the gene duplication between eS27 and eS27L, rather than just speculate on it. Also, their alignment in Figure 1A and the description about evolutionary conservation of any kind should ideally include non-tetrapod vertebrates, such as zebrafish, spotted gar, sharks, and jawless fishes, even if they cannot produce any data from laboratory experiments using these animals. It is very unbalanced that the authors invested so much on functional characterizations of the two proteins, but pay little attention to molecular phylogeny which should be the basis of the study.

We agree with this excellent suggestion and have followed these recommendations. Please see the “Essential revisions” section above, item #2, for our response.

The title of the manuscript should include the specific names of the target genes of the study.

We thank the reviewer for this suggestion and have added “Rps27 and Rps27l” to the title. We agree that this change makes the title more informative and specific.

Reviewer #3 (Recommendations for the authors):1. Figure 1A: Conclusion from Figure 1A is that the eS27 gene duplication most likely occurred through a DNA-based gene duplication event in an ancestor common to most vertebrates. However, in this figure, the alignments are mostly mammals, with only one bird and one amphibian represented. Given that, in the end, this is a paper about evolutionary biology, they should go deeper into the phylogenetic analysis. In particular, they should more deeply examine the junction between vertebrates and invertebrates. For example, in amphibians, is S27 also duplicated in salamanders? In Caecillians? And in fish, they should perform a wider analysis, from the most "recent", e.g. lungfish and coelacanths, down to sharks, hagfish and lamprey. It would also be interesting to examine a few advanced invertebrates such as sea lilies, starfish and sea urchins.

We agree with this excellent suggestion and have followed these recommendations. Please see the “Essential revisions” section above, item #2, for our response.

2. Figure 3B, closeup insert: what is the identity of the SSU protein that the eS27 amino acid residues 5, 12, and 17 about? Might this be important? At the very least, this RP should be labeled.

The ribosomal proteins surrounding Rps27 in the ribosome structure have been labeled in the inset of Figure 3B. We thank the reviewer for this suggestion to make this figure more informative.

3. I found it interesting that both liver and lactating mammary glands preferentially contain eS27L enriched ribosomes as compared to heart, brain and nulliparous mammary gland. I have two observations regarding this. First, both are organs that tend to undergo higher amounts of cell replication compared to heart, brain and NP mammary gland. This is consistent with the observation of higher association of eS27L ribosomes with cell cycle-associated mRNAs. Testing whether ribosomes are reprogrammed to contain eS27L in highly dividing versus stationary phase cultured cells would be a relatively quick and inexpensive way to test this idea. Secondly, these two organs also produce large amounts of secreted proteins as compared to the other organs. Given that the group has tagged versions of the two proteins, they could test whether or not eS27L ribosomes are preferentially associated with endoplasmic reticulum than their eS27 counterparts using cultured cells. Alternatively, they might examine whether ribosomes are reprogrammed to contain eS27L when cells are programmed to become secretory cells, e.g. compare resting B-cells to B-cells stimulated to secrete antibody (i.e. plasma cells). I note that these experiments are mere suggestions, and are not required.

We thank the reviewer for these thoughtful suggestions and agree that it remains intriguing that Rps27l-enriched tissues tend to be highly proliferative and secretory. We note that although Rps27l-ribosomes were associated with cell cycle-associated mRNAs, our analysis of cell cycle dynamics in mouse embryonic fibroblasts revealed no differences between wild-type and homogenized lines (Figure 7). However, we acknowledge that there are many other assays and conditions that could be used to characterize cell cycle parameters, which we hope to pursue in the future. Similarly, we note that our homogenized mice exhibited no signs of disease that might be expected if ER-associated translation were impaired, but acknowledge that subtle changes in ER function may not manifest in mouse models.

4. In a similar vein, rapidly duplicating and secreting cells such as liver and lactating mammary glands synthesize more protein than brain, heart and NP mammary gland cells. Perhaps the eS27L form produces ribosomes that are better tuned for speed versus accuracy. This can be tested by measuring rates of protein synthesis differences in translational accuracy in eS27 and eS27L homogenized cell lines.

We agree that this would be an interesting future direction to explore. Since Rps27/Rps27l is not positioned near the peptidyltransferase center, mRNA groove, or peptide exit tunnel of the ribosome (Figure 3B), we would consider whether the differences between the two paralogs might affect translation kinetics through longer-distance allosteric structural changes, or by differentially recruiting ribosome-associated factors.

5. Although it is clear that there are no differences between the homogenized mice and their heterologous littermates, one possible experiment would be to examine whether homogenization is disfavored over the course of multiple generations, i.e. a long term fitness experiment. However, given the extremely high costs and potentially low impact of such an experiment, I do not recommend that it be done. This question might better be performed in another, less expensive model organism such as *C. elegans* or *D. melanogaster* using a different paralogous pair of RP genes.

We agree that this would be an excellent future direction for further characterization of these paralogs and could potentially reveal subtle distinctions in protein function that were not apparent under the conditions we examined here.

6. p. 13, line 14: Sentence beginning with 'nevertheless…' is superfluous.

This sentence has been removed: “Nevertheless, to our knowledge, this is the first instance of comparing RP paralogs by endogenously epitope-tagging each paralog…”